# Thompson Sampling in Function Spaces
# via Neural Operators

**Rafael Oliveira**[*]
CSIRO's Data61
Sydney, Australia

**Xuesong Wang**
CSIRO's Data61
Sydney, Australia

**Kian Ming A. Chai**
DSO National Laboratories
Singapore

**Edwin V. Bonilla**
CSIRO's Data61
Sydney, Australia

## Abstract

We propose an extension of Thompson sampling to optimization problems over function spaces where the objective is a known functional of an unknown operator's output. We assume that queries to the operator (such as running a high-fidelity simulator or physical experiment) are costly, while functional evaluations on the operator's output are inexpensive. Our algorithm employs a sample-then-optimize approach using neural operator surrogates. This strategy avoids explicit uncertainty quantification by treating trained neural operators as approximate samples from a Gaussian process (GP) posterior. We derive regret bounds and theoretical results connecting neural operators with GPs in infinite-dimensional settings. Experiments benchmark our method against other Bayesian optimization baselines on functional optimization tasks involving partial differential equations of physical systems, demonstrating better sample efficiency and significant performance gains.

## 1 Introduction

Neural operators have established themselves as versatile models capable of learning complex, nonlinear mappings between function spaces [1]. They have demonstrated success across diverse fields, including climate science [2], materials engineering [3], and computational fluid dynamics [4]. Although their applications in supervised learning and physical system emulation are well-studied, their potential for online learning and optimization within infinite-dimensional function spaces remains relatively untapped.

In many scientific contexts, learning operators that map between function spaces naturally arises, such as the task of approximating solution operators for a partial differential equation (PDE) [1]. However, adaptive methods that efficiently query these operators to optimize functional objectives of their outputs (particularly in an active learning setting) are still underdeveloped. For example, when designing porous structures, one is often interested in optimizing how liquids flow through the structure using, e.g., Darcy flow PDEs [5], and, in the sciences, inverse problems can be solved by optimization to infer initial conditions or parameters of a physical process from observations [6, 7].

To address this gap, we propose a framework that integrates neural operator surrogates with Thompson sampling-based acquisition strategies [8] to actively optimize objectives of the form:

$$a^* \in \operatorname*{argmax}_{a \in \mathcal{A}} f(G_*(a)),$$

where $G_* : \mathcal{A} \to \mathcal{U}$ is an unknown operator between function spaces $\mathcal{A}$ and $\mathcal{U}$, and $f : \mathcal{U} \to \mathbb{R}$ is a known functional. We follow the steps of Bayesian optimization frameworks for composite functions [9, 10], which leverage knowledge of the composite structure to speed-up optimization, extending these frameworks to functional domains. Applying the theoretical results for the infinite-width limit

---

[*]Corresponding author: `rafael.dossantosdeoliveira@data61.csiro.au`

39th Conference on Neural Information Processing Systems (NeurIPS 2025).

of neural networks [11, 12], we show that a trained neural operator approximates a posterior sample from a vector-valued Gaussian process [13–15] in a sample-then-optimize approach [16]. Therefore, we are able to implement an approximate form of Thompson sampling without the need for expensive uncertainty quantification frameworks for neural operators, such as deep ensembles [17] or mixture density networks [18], and derive theoretical regret bounds on its performance. Experiments evaluate our approach on problems with classic PDE benchmarks against Bayesian optimization baselines.

## 2   Related work

**Bayesian optimization with functionals and operators.**   Bayesian optimization (BO) has been a successful approach for optimization problems involving expensive-to-evaluate black-box functions [19]. Prior work on BO in function spaces includes Bayesian Functional Optimization (BFO) [20], which uses Gaussian processes to model objectives defined over functions, focusing on scalar functionals without explicitly learning operators. Follow-up work extended the framework to include prior information about the structure of the admissible input functions [21]. Astudillo and Frazier [9] introduced the framework of composite Bayesian optimization, which was later applied by Guilhoto and Perdikaris [10] to optimization problems involving mappings from *finite-dimensional* inputs to *function-valued* outputs. Their objective was to optimize a known functional of these function-valued outputs. Our approach differs by directly working in function spaces, involving function-to-function operators. Despite the availability of GP models for function-to-function mappings [22], we are unaware of BO or GP-based bandit algorithms incorporating such models. Lastly, in the bandits literature, Tran-Thanh and Yu [23] introduced the problem of functional bandits. Despite the terminology, they deal with the problem of optimizing a known functional of the arms rewards *distribution*, similar to the setting of distributionally robust BO [24], and therefore not directly comparable to our case.

**Thompson sampling with neural networks.**   Neural Thompson Sampling (NTS) [25] employs neural networks trained via random initialization and gradient descent to approximate posterior distributions for bandit problems with scalar inputs and outputs, inspiring our use of randomized neural training for operator posterior sampling. The Sample-Then-Optimize Batch NTS (STO-BNTS) variant [16] refines this by defining acquisition functions on functionals of posterior samples, facilitating composite objective optimization. STO-BNTS extends this to batch settings using Neural Tangent Kernel (NTK) and Gaussian process surrogates, relevant for future batched active learning with neural operators. These approaches rely on the NTK theory [11], which shows that infinitely wide neural networks trained via gradient descent behave as Gaussian processes. To the best of our knowledge, this approach has not yet been extended to the case of neural network models with function-valued inputs, such as neural operators.

**Active learning for neural operators.**   Pickering et al. [17] applied deep operator networks (DeepONets) [26] to the problem of Bayesian experimental design [27]. In that framework, the goal is to select informative inputs (or designs) to reduce uncertainty about an unknown operator. To quantify uncertainty, Pickering et al. [17] used an ensemble of DeepONets and quantified uncertainty in their predictions based on the variance of the ensemble outputs. Li et al. [18] introduced multi-resolution active learning with Gaussian mixture models derived from Fourier neural operators [28]. With probabilistic outputs, mutual information can be directly quantified for active learning and Bayesian experimental design approaches. Lastly, Musekamp et al. [29] proposed a benchmark for neural operator active learning and evaluated ensemble-based models with variance-based uncertainty quantification on tasks involving forecasting. In contrast to our focus in this paper, active learning approaches are purely focused on uncertainty reduction, neglecting other optimization objectives.

## 3   Preliminaries

**Problem formulation.**   Let $\mathcal{A}$ and $\mathcal{U}$ denote two function spaces, and let $G_* : \mathcal{A} \to \mathcal{U}$ be an unknown target operator[2] between them. Consider an objective functional $f : \mathcal{U} \to \mathbb{R}$, which is

---

[2]Here, we use the term *unknown* loosely, in the sense that it is not fully implementable within the computational resources or paradigms accessible to us. For example, the target operator can be a simulator in a high-performance computing facility which we have limited access to.

assumed known and cheap to evaluate. Given a compact search space $\mathcal{S} \subset \mathcal{A}$, we aim to solve:[3]

$$a^* \in \underset{a \in \mathcal{S}}{\operatorname{argmax}} f(G_*(a)), \tag{1}$$

while $G_*$ is only accessible via expensive oracle queries: for a chosen $a$, we observe a function-valued output $y = HG_*(a) + \xi$, where $H : \mathcal{U} \to \mathcal{Y}$ represents an observation operator, typically the discretization on a grid, with $\mathcal{Y}$ being a (usually finite-dimensional) Hilbert space, and $\xi \sim \mathcal{N}(0, \Sigma)$ is observation noise, assumed independent and identically distributed (i.i.d.) across queries. The algorithm is allowed to query the oracle with any function $a \in \mathcal{S}$ for up to a budget of $N$ queries. For this paper, we focus on problems with finite search space $|\mathcal{S}| < \infty$, though the framework is general.

**Neural operators.** A neural operator is a specialized neural network architecture modeling operators $G : \mathcal{A} \to \mathcal{U}$ between function spaces $\mathcal{A}$ and $\mathcal{U}$ [1]. Assume $\mathcal{A} \subset \mathcal{C}(\mathcal{X}, \mathbb{R}^{d_a})$ and $\mathcal{U} \subset \mathcal{C}(\mathcal{Z}, \mathbb{R}^{d_u})$, where $\mathcal{C}(\mathcal{S}, \mathcal{S}')$ denotes the space of continuous functions between sets $\mathcal{S}$ and $\mathcal{S}'$. Given an input function $a \in \mathcal{A}$, a neural operator $G_{\boldsymbol{\theta}}$ performs a sequence of transformations $a =: u_1 \mapsto \cdots \mapsto u_{L-1} \mapsto u_L$ through $L$ layers of neural networks, where $u_l : \mathcal{X}_l \to \mathbb{R}^{d_l}$ is a continuous function for each layer $l \in \{1, \ldots, L\}$, and $\mathcal{X}_L := \mathcal{Z}$ is the domain of the output functions and $d_L := d_u$. In one of its general formulations, for a given layer $l \in \{1, \ldots, L\}$, the result of the transform (or update) at any $x \in \mathcal{X}_{l+1}$ can be described as:

$$u_1(x) := a(x)$$
$$u_{l+1}(x) := \alpha_l \left( \int_{\mathcal{X}_l} \mathbf{R}_l(x, x', u_l(\Pi_l(x)), u_l(x')) \, u_l(x') \, \mathrm{d}\nu_l(x') + \mathbf{W}_l \, u_l(\Pi_l(x)) + b_l(x) \right) \tag{2}$$
$$G_{\boldsymbol{\theta}}(a)(z) := u_L(z),$$

where $\Pi_l : \mathcal{X}_{l+1} \to \mathcal{X}_l$ is a fixed mapping, $\alpha_l : \mathbb{R} \to \mathbb{R}$ denotes an activation function applied elementwise, $\mathbf{R}_l : \mathcal{X}_{t+1} \times \mathcal{X}_t \times \mathbb{R}^{d_l} \times \mathbb{R}^{d_l} \to \mathbb{R}^{d_{t+1} \times d_t}$ defines a (possibly nonlinear or positive-semidefinite) kernel integral operator with respect to a measure $\nu_l$ on $\mathcal{X}_l$, $\mathbf{W}_l \in \mathbb{R}^{d_{l+1} \times d_l}$ is a weight matrix, and $b_l : \mathcal{X}_{l+1} \to \mathbb{R}^{d_{l+1}}$ is a bias function. We denote by $\boldsymbol{\theta}$ the collection of all learnable parameters of the neural operator: the weights matrices $\mathbf{W}_l$, the parameters of the bias functions $b_l$ and the matrix-valued kernels $\mathbf{R}_l$, for all layers $l \in \{1, \ldots, L\}$. Variations to the formulation above correspond to various neural operator architectures based on low-rank kernel approximations, graph structures, Fourier transforms, etc. [1].

**Vector-valued Gaussian processes.** Vector-valued Gaussian processes extend scalar GPs [13] to the case of vector-valued functions [14]. Let $\mathcal{A}$ be an arbitrary domain, and let $\mathcal{U}$ be a Hilbert space representing a codomain. We consider the case where both the domain $\mathcal{A}$ and codomain $\mathcal{U}$ might be infinite-dimensional vector spaces, which leads to GPs whose realizations are operators $G_* : \mathcal{A} \to \mathcal{U}$ [15]. To simplify our exposition, we assume that $\mathcal{U}$ is a separable Hilbert space, though the theoretical framework is general enough to be extended to arbitrary Banach spaces [30]. A vector-valued Gaussian process $G_* \sim \mathcal{GP}(\widehat{G}, K)$ on $\mathcal{A}$ is fully specified by a mean operator $\widehat{G} : \mathcal{A} \to \mathcal{U}$ and a positive-semidefinite operator-valued covariance function $K : \mathcal{A} \times \mathcal{A} \to \mathcal{L}(\mathcal{U})$, where $\mathcal{L}(\mathcal{U})$ denotes the space of bounded linear operators on $\mathcal{U}$. Formally, given any $a, a' \in \mathcal{A}$ and any $u, u' \in \mathcal{U}$, it follows that:

$$\mathbb{E}[G_*(a)] = \widehat{G}(a), \tag{3}$$
$$\operatorname{Cov}(\langle G_*(a), u \rangle, \langle G_*(a'), u' \rangle) = \langle u, K(a, a')u' \rangle, \tag{4}$$

where $\langle \cdot, \cdot \rangle$ denotes the inner product and $\operatorname{Cov}(\cdot, \cdot)$ stands for the covariance between scalar variables. Assume we are given a set of observations $\mathcal{D}_t := \{(a_i, y_i)\}_{i=1}^t \subset \mathcal{A} \times \mathcal{U}$, where $y_i = G_*(a_i) + \xi_i$, and $\xi_i \sim \mathcal{N}(0, \Sigma)$ corresponds to Gaussian noise with covariance operator $\Sigma \in \mathcal{L}(\mathcal{U})$. The posterior mean and covariance can then be defined by the following recursive relations:

$$\widehat{G}_t(a) = \widehat{G}_{t-1}(a) + K_{t-1}(a, a_t)(K_{t-1}(a_t, a_t) + \Sigma)^{-1}(y_t - \widehat{G}_{t-1}(a_t)) \tag{5}$$
$$K_t(a, a') = K_{t-1}(a, a') - K_{t-1}(a, a_t)(K_{t-1}(a_t, a_t) + \Sigma)^{-1}K_{t-1}(a_t, a') \tag{6}$$

---

[3]We use "$\in \operatorname{argmax}$" acknowledging that the problem may have multiple global optima, forming a set of global optimizers. Whenever we assume a unique minimizer, we will use the equality symbol "=", instead.

| **Algorithm 1:** GP-TS | **Algorithm 2:** NOTS (ours) |
|---|---|
| **Input:** Search space $\mathcal{S}$, initial data $\mathcal{D}_0$ | **Input:** Search space $\mathcal{S}$, initial data $\mathcal{D}_0$ |
| **for** $t \in \{1, \ldots, T\}$ **do** | **for** $t = 1, \ldots, T$ **do** |
| $\quad$ Sample $g_t \sim \mathcal{GP}(\mu_{t-1}, k_{t-1})$ | $\quad \boldsymbol{\theta}_t = \arg\min_{\boldsymbol{\theta}} \ell_t(\boldsymbol{\theta}), \;\; \boldsymbol{\theta}_{t,0} \sim \mathcal{N}(\mathbf{0}, \boldsymbol{\Sigma}_0)$ |
| $\quad$ Select $x_t \in \arg\max_{x \in \mathcal{X}} g_t(x)$ | $\quad a_t \in \arg\max_{a \in \mathcal{S}} f(G_{\boldsymbol{\theta}_t}(a))$ |
| $\quad$ Query $y_t = f(x_t) + \epsilon_t$ | $\quad y_t = G_*(a_t) + \xi_t$ |
| $\quad$ Update $\mathcal{D}_t = \mathcal{D}_{t-1} \cup \{x_t, y_t\}$ | $\quad \mathcal{D}_t = \mathcal{D}_{t-1} \cup \{a_t, y_t\}$ |

for any $a, a' \in \mathcal{A}$, and $t \in \mathbb{N}$, which are an extension of the same recursions from the scalar-valued case [31, App. F] to the case of vector-valued processes. Such definition arises from sequentially conditioning the GP posterior on each observation, starting from the prior with $\widehat{G}_0 := \widehat{G}$ and $K_0 := K$. This recursion leads to the same matrix-based definitions of the usual GP posterior equations [13], but in our case it avoids complications with the resulting higher-order tensors that arise when kernels are operator-valued.

**Thompson sampling.** Thompson sampling (TS) is a relatively simple randomized strategy for sequential decision making under uncertainty, which has found many successes in the Bayesian optimization and multi-armed bandits literature [8, 25, 32, 33]. When applied to optimization problems, the core idea of TS is to query an objective function $f$ at points $x_t$ sampled from the probability distribution of the optimum location $x^* \in \arg\max_{x \in \mathcal{X}} f(x)$ given the observations $\mathcal{D}_{t-1} := \{x_i, y_i\}_{i=1}^{t-1}$. To do so, the objective function is modeled as sample from a Bayesian probabilistic model, which is typically a linear model [8] or a GP [33], and then TS samples realizations $g_t$ of the objective from the model's posterior $p(f|\mathcal{D}_{t-1})$. A point $x_t$ which maximizes a sampled function $g_t$ then corresponds to a sample from the posterior distribution over the optimum $p(x^*|\mathcal{D}_{t-1})$. The procedure is summarized in Algorithm 1 for the case of a GP. Under mild assumptions, TS is known to produce a sequence of candidates $x_t$ such that $f(x_t)$ asymptotically converges to $f(x^*)$ [33, 34].

## 4 Neural operator Thompson sampling

We propose a Thompson sampling algorithm for the optimization of functionals of unknown operators in the setting of Eq. 1. Instead of relying on extensions of traditional probabilistic methods to operator modeling, our method applies flexible and scalable neural operators as surrogates $G_t$, training them to approximate posterior samples over the true operator $G_*$ conditioned on data. The method is designed to efficiently explore the search space while balancing the exploration-exploitation trade-off.

### 4.1 Approximate posterior sampling

Given data $\mathcal{D}_t = \{(a_i, y_i)\}_{i=1}^t$, we train a neural operator $G_{\boldsymbol{\theta}}$ with parameters $\boldsymbol{\theta}_t$ that minimize:

$$\ell_t(\boldsymbol{\theta}) := \sum_{j=1}^{t-1} \|y_j - HG_{\boldsymbol{\theta}}(a_j)\|^2 + \lambda\|\boldsymbol{\theta}\|^2, \tag{7}$$

where $\|\cdot\|$ represents the norm in the underlying vector space, and $\lambda > 0$ is a regularization factor which relates to the noise process $\xi$ [35]. We minimize $\ell_t$ via gradient descent starting from $\boldsymbol{\theta}_{t,0} \sim \mathcal{N}(0, \boldsymbol{\Sigma}_0)$, where $\boldsymbol{\Sigma}_0$ is a diagonal matrix following Kaiming He [36] or LeCun initialization [37], which scale each layer's weights initialization variance according to the width of the previous layer. By an extension of standard results on the infinite-width limit of neural networks to the neural operator setting, we can show that the trained neural operator approximates a posterior sample from a vector-valued GP when, e.g., we train only the last linear layer (see App. C.4), which in turn guarantees regret bounds (Sec. 5). The prior over $G_*$ is implicitly defined as the vector-valued Gaussian process given by the conjugate kernel [38, 39] associated with the neural operator architecture and the weights initialization distribution. Lastly, we note that, in practice, observations are discretized over a finite grid or other finite-dimensional representation [1], so that the observation space is $\mathcal{Y} \subseteq \mathbb{R}^m$ and the difference norms in Eq. 7 reduce to Euclidean distances.

## 4.2 Thompson sampling algorithm

In Algorithm 2, we present the Neural Operator Thompson Sampling (NOTS) algorithm for the optimization of problem-dependent functionals of black-box operators. The algorithm operates sequentially over $T$ iterations similar to standard GP-TS (Algorithm 1). To sample a realization from the neural operator posterior, each iteration begins with the random initialization of the parameters of a neural operator that serves as a surrogate model for the true unknown operator. At each iteration, the neural operator model is trained according to Section 4.1, minimizing a regularized least-squares loss based on the currently available data, yielding an approximate sample $G_t := G_{\boldsymbol{\theta}_t}$ from the true operator posterior $p(G_* | \mathcal{D}_{t-1})$. The next step involves selecting the input for querying the oracle by maximizing the value of the objective functional $f$ over the neural operator's predictions $G_t(a)$. Finally, the algorithm runs the potentially expensive step of querying the true operator $G_*$ with the selected input function $a_t$, which may involve a complex simulation or physical experiment, and updates the dataset with the new (noisy) observation $y_t$. This process repeats for up to $T$ iterations, producing a sequence of function-valued queries $a_t$ that approximates the true optimum $a^*$ (1).

**Computational cost.** Each iteration of NOTS incurs a linear computational cost of $\mathcal{O}(t)$ due to the retraining of the neural operator model, which can be further reduced by use of minibatch stochastic gradient descent. The reinitialization with randomized weights followed by retraining is what ensures that we have a new approximate posterior sample for TS conditioned on the available data at every iteration. Compared to a more traditional GP-based approach, which applied to our setting would incur a $\mathcal{O}(t^3)$ cost per step due to the inversion of a covariance matrix of $t$ data points, we achieve a much more computationally efficient and scalable algorithm, despite the cost of retraining the model.

## 5 Theoretical results

In this section, we establish the theoretical foundation of our proposed method. We show how a randomly initialized neural operator approximates a GP in the infinite-width limit through the use of the conjugate kernel, also known as the NNGP kernel [38–42], under certain assumptions. This allows us to extend existing results for GP Thompson Sampling (GP-TS) [33] to our setting.

### 5.1 Neural operator abstraction

A neural operator models nonlinear operators $G : \mathcal{A} \to \mathcal{U}$ between possibly infinite-dimensional function spaces $\mathcal{A}$ and $\mathcal{U}$. Current results in NTK [11] and GP limits for neural networks [12] do not immediately apply to this setting, as they rely on finite-dimensional domains. However, we can leverage an abstraction for neural operator architectures which sees their layers as maps over finite-dimensional inputs [43], which result from truncations to make the modeling problem tractable.

Considering a neural operator with a *single* hidden layer, let $M \in \mathbb{N}$ represent the layer's width, $A_{\mathbf{R}} : \mathcal{A} \to \mathcal{C}(\mathcal{Z}, \mathbb{R}^{d_{\mathbf{R}}})$ denote a (fixed) continuous operator, and $b_0 : \mathcal{Z} \to \mathbb{R}^{d_b}$ denote a (fixed) continuous function. For simplicity, we will assume scalar-valued output functions with $d_u = 1$. In general, with a single hidden layer, the model described in Eq. 2 can be rewritten as:

$$G_{\boldsymbol{\theta}}(a)(z) = \mathbf{w}_o^{\mathsf{T}} \alpha \left( \mathbf{W}_{\mathbf{R}} A_{\mathbf{R}}(a)(z) + \mathbf{W}_u a(\Pi_0(z)) + \mathbf{W}_b b_0(z) \right), \quad z \in \mathcal{Z}, \qquad (8)$$

where $\boldsymbol{\theta} := \mathrm{vec}(\mathbf{w}_o, \mathbf{W}_{\mathbf{R}}, \mathbf{W}_u, \mathbf{W}_b) \in \mathbb{R}^{M(1+d_{\mathbf{R}}+d_a+d_b)} =: \mathcal{W}$ represents the model's flattened parameters. The finite weight matrix $\mathbf{W}_{\mathbf{R}}$ representing the kernel convolution integral arises as a result of truncations required in the practical implementation of neural operators (e.g., a finite number of Fourier modes or quadrature points). With this formulation, one can recover most popular neural operator architectures [43]. In Appendix B, we discuss how Fourier neural operators [28] fit under this formulation, though the latter is general enough to incorporate other cases. We also highlight that neural operators possess universal approximation properties [44], given sufficient data and computational resources, despite the inherent low-rank approximations in their architecture.

### 5.2 Infinite-width limit of neural operators

With the construction in Eq. 8, we can simply see the result of a neural operator layer when evaluated at a fixed $z \in \mathcal{Z}$ equivalently as a $M$-width feedforward neural network:

$$G_{\boldsymbol{\theta}}(a)(z) = h_{\boldsymbol{\theta}}(\mathbf{v}_z(a)) := \mathbf{w}_o^{\mathsf{T}} \alpha(\mathbf{W} \mathbf{v}_z(a)), \qquad (9)$$

where the input is given by $\mathbf{v}_z(a) := [A_{\mathbf{R}}(a)(z),\ a(\Pi_0(z)),\ b_0(z)] \in \mathcal{V}$, and $\mathcal{V} := \mathbb{R}^{d_{\mathbf{R}}+d_a+d_b}$.

**Conjugate kernel.** We can now derive infinite-width limits. The conjugate kernel describes the distribution of the untrained neural network $h_{\boldsymbol{\theta}} : \mathcal{V} \to \mathbb{R}$ under Gaussian weights initialization, whose infinite-width limit yields a Gaussian process [38, 40]. Formally, the conjugate kernel is defined as:

$$k_h(\mathbf{v}, \mathbf{v}') := \lim_{M \to \infty} \mathbb{E}_{\boldsymbol{\theta}_0 \sim \mathcal{N}(\mathbf{0}, \boldsymbol{\Sigma}_0)}[h_{\boldsymbol{\theta}_0}(\mathbf{v}) h_{\boldsymbol{\theta}_0}(\mathbf{v}')], \quad \mathbf{v}, \mathbf{v}' \in \mathcal{V}. \tag{10}$$

Further background on the conjugate kernel and the NTK can be found in Appendix A.

Since the composition of the map $\mathcal{A} \times \mathcal{Z} \ni (a, z) \mapsto \mathbf{v}_z(a) \in \mathcal{V}$ with a kernel on $\mathcal{V}$ yields a kernel on $\mathcal{A} \times \mathcal{Z}$ [45, Lem. 4.3], the conjugate kernel of $G_{\boldsymbol{\theta}}$ is determined by:

$$k_G(a, z, a', z') := k_h(\mathbf{v}_z(a), \mathbf{v}_{z'}(a')), \quad a, a' \in \mathcal{A}, \quad z, z' \in \mathcal{Z}, \tag{11}$$

where $k_h$ is the conjugate kernel of the neural network $h_{\boldsymbol{\theta}}$. Such a kernel defines a covariance function for a GP over the space of operators mapping $\mathcal{A}$ to $\mathcal{U}$. Assume $\mathcal{U} \subset \mathcal{L}^2(\nu)$ is a closed subspace of the space of functions which are square integrable with respect to a $\sigma$-finite Borel measure on $\mathcal{Z}$, and let $\mathcal{L}(\mathcal{U})$ denote the space of linear operators on $\mathcal{U}$. The following then defines a positive-semidefinite operator-valued kernel $K_G : \mathcal{A} \times \mathcal{A} \to \mathcal{L}(\mathcal{U})$:

$$(K_G(a, a')u)(z) = \int_{\mathcal{Z}} k_G(a, z, a', z')u(z')\, \mathrm{d}\nu(z'), \tag{12}$$

for any $u \in \mathcal{U}$, $a, a' \in \mathcal{A}$ and $z \in \mathcal{Z}$. Hence, we can state the following result, whose proof can be found in Appendix C.2.

**Proposition 1.** *Let $G_{\boldsymbol{\theta}} : \mathcal{A} \to \mathcal{U}$ be a neural operator with a single hidden layer, where $\mathcal{U} \subseteq \mathcal{L}^2(\nu)$ is closed, and $\nu$ is a finite Borel measure on $\mathcal{Z}$. Assume $\mathbf{w}_o \sim \mathcal{N}(\mathbf{0}, \sigma_{\boldsymbol{\theta}}^2 \mathbf{I})$, for $\sigma_{\boldsymbol{\theta}}^2 > 0$ such that $\sigma_{\boldsymbol{\theta}}^2 \propto 1/M$, while the remaining parameters have their entries sampled from a fixed normal distribution. Then, as $M \to \infty$, on every compact subset of $\mathcal{A}$, the neural operator converges in distribution to a zero-mean vector-valued Gaussian process with operator-valued covariance function given by:*

$$\lim_{M \to \infty} \mathbb{E}_{\boldsymbol{\theta} \sim \mathcal{N}(\mathbf{0}, \boldsymbol{\Sigma}_0)}[G_{\boldsymbol{\theta}}(a) \otimes G_{\boldsymbol{\theta}}(a')] = K_G(a, a'), \quad a, a' \in \mathcal{A},$$

*where $K_G : \mathcal{A} \times \mathcal{A} \to \mathcal{L}(\mathcal{U})$ is defined in Eq. 12, and $\otimes$ denotes the outer product.*

### 5.3 Bayesian cumulative regret bounds

**Bayesian regret.** We analyze the performance of a sequential decision-making algorithm via its Bayesian cumulative regret. An algorithm's instant regret for querying $a_t \in \mathcal{A}$ at iteration $t \geq 1$ is:

$$r_t := f(G_*(a^*)) - f(G_*(a_t)) \tag{13}$$

where $a^*$ is defined in Eq. 1. The Bayesian cumulative regret after $T$ iterations is then defined as:

$$R_T := \mathbb{E}\left[\sum_{t=1}^{T} r_t\right], \tag{14}$$

where the expectation is over all sources of randomness affecting the decision-making process, i.e., the prior for $G_*$ and the observation noise. If the algorithm achieves sub-linear cumulative regret, its simple regret asymptotically vanishes, as $\lim_{T \to \infty} \mathbb{E}\left[\min_{t \in \{1,...,T\}} r_t\right] \leq \lim_{T \to \infty} \frac{1}{T} R_T$, leading the algorithm's queries $a_t$ to eventually approach the true optimum $a^*$.

**Regularity assumptions.** For our analysis, we assume $\mathcal{U} \subseteq \mathcal{L}^2(\nu)$ is a closed subspace of the Hilbert space $\mathcal{L}^2(\nu)$ of square-integrable $\nu$-measurable functions, for a given finite Borel measure $\nu$ on a compact domain $\mathcal{Z}$. We will assume the search space $\mathcal{S} \subset \mathcal{A}$ is finite. The true operator $G_* : \mathcal{A} \to \mathcal{U}$ will be assumed to be a sample from a vector-valued Gaussian process $G_* \sim \mathcal{GP}(0, K)$, where the operator-valued kernel $K : \mathcal{A} \times \mathcal{A} \to \mathcal{L}(\mathcal{U})$ is given by the neural operator's infinite-width limit in Proposition 1. Observations $y = HG_*(a) + \xi$ are assumed to be corrupted by i.i.d. zero-mean Gaussian noise, $\xi \sim \mathcal{N}(0, \Sigma)$, where $\Sigma$ is positive definite on $\mathcal{Y} \subseteq \mathbb{R}^m$.

We adapt state-of-the-art regret bounds for GP-TS [33] to an exact version of NOTS. To do so, we first observe that, for a linear functional $f \in \mathcal{L}(\mathcal{U}, \mathbb{R})$, the composition with a Gaussian random operator $G_* \sim \mathcal{GP}(\widehat{G}, K)$ yields a scalar-valued GP, i.e., $f \circ G_* \sim \mathcal{GP}(f \circ \widehat{G}, f^{\mathsf{T}} K f)$, where $f^{\mathsf{T}} K f : (a, a') \mapsto f(K(a, a')f)$. We can then extend GP-TS regret bounds to the case of operators.

**Proposition 2.** *Let $f : \mathcal{U} \to \mathbb{R}$ be a bounded linear functional such that $f = \tilde{f} \circ H$, where $\tilde{f} : \mathcal{Y} \to \mathbb{R}$ is linear, and $G_* \sim \mathcal{GP}(0, K)$. Consider a sequential algorithm selecting $a_t \in \arg\max_{a \in \mathcal{S}} f(G_t(a))$ and observing $y_t = HG_*(a_t) + \xi_t$, where $G_t \overset{d}{=} G_* | \mathcal{D}_t$, and $\xi_t \sim \mathcal{N}(0, \lambda I)$, for $t \in \{1, \dots, T\}$. Then, this algorithm's expected cumulative regret is such that:*

$$R_T \in \tilde{\mathcal{O}}(\sqrt{T}), \tag{15}$$

*where $\tilde{\mathcal{O}}(\cdot)$ suppresses logarithmic factors of the $\mathcal{O}(\cdot)$ asymptotic rate.*

This result shows that NOTS can achieve sublinear cumulative regret in the infinite-width limit with an exact GP posterior sample. The result connects existing GP-TS guarantees to NOTS, and it differs from existing guarantees for other neural network based Thompson sampling algorithms [16, 25], which explored the scalar case and a frequentist setting (i.e., the objective function being a fixed element of the reproducing kernel Hilbert space defined by the network's neural tangent kernel). In the Bayesian setting, there is also no need for a time-dependent regularization parameter [16], allowing for a simpler implementation. Yet we note that Proposition 2 concerns the exact GP case. However, Proposition 1 ensures that a single-hidden-layer randomly initialized neural operator follows a GP in the infinite-width limit, and we show in the appendix that training the last layer via gradient descent approximates a posterior sample, as in previous results for conventional neural networks [12, App. D]. Appendix C presents proofs and further discussions on limitations and extensions, and a validation experiment can be found in Appendix E.

## 6 Experiments

We evaluate our NOTS algorithm on two popular PDE benchmark problems: Darcy flow and a shallow water model. Our results are compared against a series of representative Bayesian optimization and neural Thompson sampling baselines. More details about our implementations and further experiment details can be found in Appendix D. Code for our experiments will be made available online.[4]

### 6.1 Algorithms

We compare NOTS against a series of GP-based and neural network BO algorithms modeling directly the mapping from function-valued inputs $a \in \mathcal{A}$ (discretized over regular grid) to the scalar-valued functional evaluations $f(G_*(a))$, besides a trivial random search (RS) baseline. NOTS is implemented with standard and spherical FNOs [46], following default library settings for these PDEs [47]. We first implemented BO with a 3-layer infinite-width ReLU Bayesian neural network (BNN) model, represented as a GP with the corresponding conjugate kernel. According to Li et al. [48], these models can achieve optimal performance in high-dimensional settings when compared to other BNN methods. Two versions of this framework are in our experiments, one with log-expected improvement, given its well established competitive performance [49], simply denoted as "BO" in our plots, and one with Thompson sampling (GP-TS) [34]. As our experiments are over finite domains, sampling from a scalar GP boils down to sampling from a multivariate normal distribution. Next, we evaluated a version of Bayesian functional optimization (BFO) by encoding input functions in a reproducing kernel Hilbert space (RKHS) via their minimum-norm interpolant and using a squared-exponential kernel over functions which takes advantage of the RKHS structure as in the original BFO [20]. Lastly, we evaluated sample-then-optimize neural Thompson sampling (STO-NTS), training a 2-layer 256-width fully connected neural network with a regularized least-squares loss [16].

### 6.2 PDE benchmarks

**Darcy flow.** Darcy flow models fluid pressure in a porous medium [28], with applications in contaminant control, leakage reduction, and filtration design. In our setting, the input $a \in \mathcal{C}((0, 1)^2, \mathbb{R}_+)$ is the medium's permeability on a Dirichlet boundary, and the operator $G_*$ maps $a$ to the pressure field $u \in \mathcal{C}((0, 1)^2, \mathbb{R})$. To train $G_\theta$, we generate 1,000 input–output pairs via a finite-difference solver at $16 \times 16$ resolution. Two materials are considered, leading to a binary grid for $a$ and a continuum of pressure values for each $u$ grid cell. More details are in Li et al. [28] and Appendix D.

---

[4]Code repository: https://github.com/csiro-funml/nots

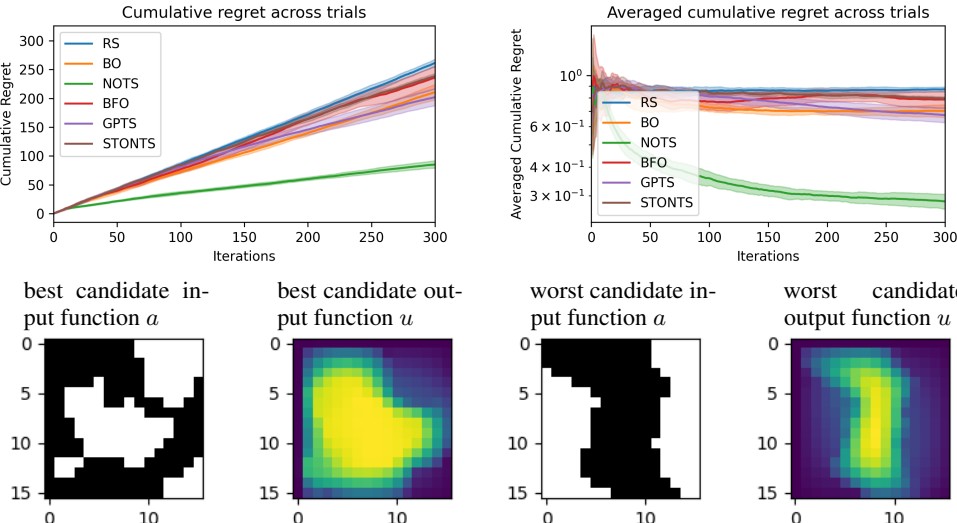

Figure 1: Darcy flow rate optimization. Overlay of cumulative regret (top left) and its average (top right) metrics across trials for the negative total flow rates case in the Darcy flow problem. The shaded areas correspond to one standard deviation across 10 trials. The corresponding input-output functions that achieved the best and worst flow rates are presented (bottom). White regions $a(x) = 1$ means fully open permeability and black regions $a(x) = 0$ represents impermeable pore material. The output function suggests pressure field where brighter color indicates higher pressure.

**Shallow water modeling.** Shallow water models capture the time evolution of fluid mass and discharge on a rotating sphere [46]. The input $a \in \mathcal{C}(\mathbb{S}^2 \times \{t = 0\}, \mathbb{R}^3)$ represents the initial geopotential depth and two velocity components, while the output $u \in \mathcal{C}(\mathbb{S}^2 \times \{t = \tau\}, \mathbb{R}^3)$ gives the state at time $t = \tau$. We train $G_\theta$ on 200 random initial conditions on a $32 \times 64$ equiangular grid, using a 1,200 s timestep to simulate up to $\tau = 6$ hours.

### 6.3 Optimization functionals

We introduce several optimization functionals that are problem-dependent and clarify their physical meaning in the context of the benchmark problems. As we aim to solve a maximization problem, physical quantities to be minimized are defined with a negative sign. The first three functionals were applied to the Darcy flow problem and the last one to shallow water modeling. Note that in both cases, we have the same domain for the PDE solutions $u$ and input functions $a$, i.e., $\mathcal{Z} = \mathcal{X}$.

**Negative total flow rates [50]** $f(u, a) = -\int_{\partial \mathcal{X}} a(x)(\nabla u(x) \cdot n) dx$. Here $\partial \mathcal{X}$ is the boundary of the domain and $n$ is the outward pointing unit normal vector of the boundary. This functional integrates the volumetric flux $-a(x)\nabla u(x)$ along the boundary, which corresponds to the total flow rate of the fluid. Such an objective can be optimized for leakage reduction and contaminant control.

**Negative total pressure [51]** $f(u) = -\frac{1}{2}\int_{\mathcal{X}} |u(x)| dx$. This objective computes the total fluid pressure over the domain in the Darcy flow system.

**Negative total potential energy** $f(u, a) = -\int_{\mathcal{X}} a(x)\|\nabla u(x)\|^2 \, dx + \int_{\mathcal{X}} s(x)u(x) \, dx$. This functional quantifies the system's total potential energy, balancing the energy dissipated by fluid friction (the first term) against the potential energy supplied by the uniform fluid source (the second term, where $s = 1$ is assumed). The minimizer $a^*$, therefore, consists of the most hydrodynamically efficient design for the given flow constraints.

**Inverse problem** $f(u) = -\frac{1}{2}\|u - u_\tau\|^2$. $u_\tau$ represents the ground truth solution. This objective is specific to shallow water modeling, as we aim to find the initial condition $a$ that generates $u_\tau$ at time $\tau$, which is also a simplification of the assimilation objective in weather forecasting [52, 53].

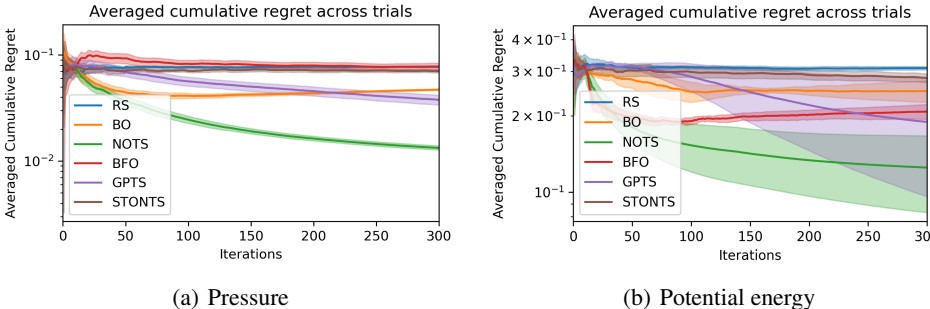

(a) Pressure          (b) Potential energy

Figure 2: Darcy flow pressure (a) and potential energy (b) optimization problems averaged cumulative regret. The shaded areas correspond to one standard deviation across 10 trials.

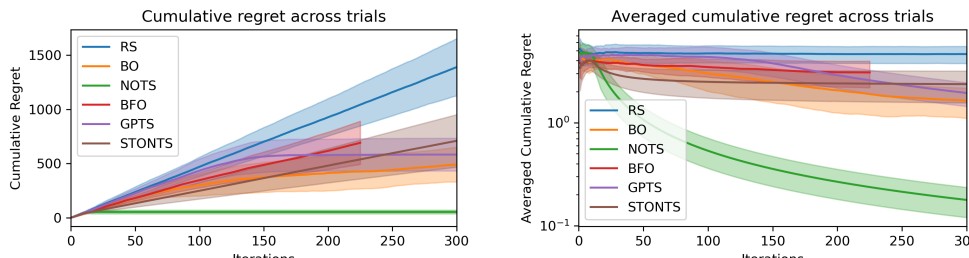

Figure 3: Shallow water inverse problem. Overlay of cumulative regret (left) and its average (right) metrics across trials for the inverse problem in the shallow water data. The shaded areas correspond to one standard deviation across 10 trials.

## 6.4 Results

Our results are presented in Figure 1 to 3, comparing the cumulative regret of NOTS against the baselines on different settings of PDE problems and functional objectives. Results are summarized in Table 1 with the final average regret, i.e., $\frac{R_T}{T}$, of each method across the different problems.

In Figure 1, we present our results for the flow rate optimization problem in the Darcy flow PDE benchmark. The results clearly show that GP-based BO methods struggle in this high-dimensional setting, while NOTS (ours) is able to consistently find optimal solutions. As described in Section 6.2, input functions $a \in \mathcal{A}$ for Darcy flow are binary masks representing two materials of different permeability which are discretized over a 2D grid of 16-by-16 sampling locations. Hence, when applied to standard GP-based BO methods, the inputs correspond to 256-dimensional vectors, which can be quite high-dimensional for standard GPs. The optimization results of the input and output functions also show the effectiveness of our approach. In the case of the "best candidate" which achieves the lowest total flow rate, the input function shows large contiguous impermeable regions that block fluid outflow and thus generate high interior pressure which can be treated as an ideal design for leakage control. In contrast, the "worst candidate" exhibits the highest total flow rates. It has smooth, boundary-connected permeable zones allowing fluid to escape effortlessly. Lastly, figures 2(a) and 2(b) show the results on optimizing pressure and potential energy on Darcy flow. On these functionals, BO and GP-TS can achieve a better performance, recalling their use of the infinite-width BNN kernel, which has shown good performance on high-dimensional problems [48]. Yet, we can see significant performance improvements from NOTS with respect to all baselines.

Figure 3 shows our results for the inverse problem on the shallow water PDE benchmark. This setting involves higher dimensional discretized inputs (6144-dimensional when flattened), leading to an extremely challenging problem for GP approaches. In particular, the evaluation of the functional inputs kernel is too computationally intensive for BFO, leading it to crash before 250 iterations are completed. We believe that STO-NTS's low performance is due to architectural limitations, as it uses a simple fully connected network, which leads to a need for higher amounts of data (i.e.,

Table 1: Results summary: Final average regret of each method and its standard deviation.

| Method | Darcy flow rates | Darcy flow energy | Darcy flow pressure | Shallow water |
|--------|------------------|-------------------|---------------------|---------------|
| RS | $0.872 \pm 0.022$ | $0.309 \pm 0.005$ | $0.077 \pm 0.001$ | $4.632 \pm 0.876$ |
| BO | $0.703 \pm 0.045$ | $0.251 \pm 0.024$ | $0.047 \pm 0.001$ | $1.639 \pm 0.532$ |
| BFO | $0.788 \pm 0.066$ | $0.208 \pm 0.014$ | $0.078 \pm 0.006$ | $3.076 \pm 0.886$ |
| GP-TS | $0.674 \pm 0.050$ | $0.189 \pm 0.093$ | $0.038 \pm 0.004$ | $1.942 \pm 0.502$ |
| STO-NTS | $0.068 \pm 0.002$ | $0.282 \pm 0.011$ | $0.068 \pm 0.002$ | $2.329 \pm 0.800$ |
| NOTS | $\mathbf{0.012 \pm 0.001}$ | $\mathbf{0.125 \pm 0.042}$ | $\mathbf{0.012 \pm 0.001}$ | $\mathbf{0.134 \pm 0.043}$ |

more iterations). NOTS, however, is able to learn the underlying physics of the problem to aid its predictions, leading to a more efficient exploration and higher performance.

# 7 Conclusion

We have developed Neural operator Thompson sampling (NOTS) for optimization problems in function spaces and shown that it provides significant performance gains in encoding the compositional structure of problems involving black-box operators, such as complex physics simulators or real physical processes. NOTS also comes equipped with theoretical guarantees, connecting the existing literature on Thompson sampling to this novel setting involving neural operators.

**Discussion.** We have shown empirically that using neural operators as surrogates for Thompson sampling can be effective without the need for expensive uncertainty quantification schemes by relying on theoretical results for infinitely wide deep neural networks and their connection with Gaussian processes. Neural operators have allowed for effective representation learning which scales to very high-dimensional settings, where traditional bandits and Bayesian optimization algorithms would struggle. Although GPs typically perform well on Bayesian modeling tasks with low volumes of data, the functional optimization problems we considered have high-dimensional data as both inputs and outputs, rendering the application of traditional multi-output GP models challenging. The basic computational complexity of inference with a vector-valued GP model scales cubically with both the number of data points and the number of output coordinates [14]. For the shallow water PDE, for example, both inputs and outputs lie in a 6144-dimensional space. With 300 iterations, a multi-output GP would have to invert a kernel matrix over more than 1 million data points towards the last iterations. Hence, without specialized kernels and computationally efficient approximations, a traditional GP approach would be unsuitable due to the very large number of outputs. In contrast, neural operators are specially designed to deal with function-valued input and output data, typically over spatial domains, with linearly scaling computational complexity. Therefore, NOTS can better scale to accommodate longer runs or extensions to batched evaluations than a GP approach, even though we limited experiments to 300 iterations to allow for comparisons against GP baselines.

**Limitations and future work.** We note that our current results are focused on the case of finite search spaces and well specified models, which provide a first step towards more general use cases. An extension to continuous domain could, for example, parameterize the set of input functions and optimize such parametric representation or tractable nonparametric extensions [20, 21], which might be application specific. Our theoretical analysis only considered the case of a neural operator with a single hidden layer, despite the multi-layer setting in our experiments. These and other limitations are further discussed in Appendix F. As future work, we plan to investigate the generalization of our results to more general settings, such as continuous domains and batched evaluations. Lastly, we note that NOTS also offers a framework for task-to-task amortization and few-shot learning, as operator learning data can be reused across tasks with different objective functionals.

# Acknowledgments and Disclosure of Funding

This research was carried out solely using CSIRO's resources. Chai contributed while on sabbatical leave visiting the Machine Learning and Data Science Unit at Okinawa Institute of Science and Technology, and the Department of Statistics in the University of Oxford. This project was supported by resources and expertise provided by CSIRO IMT Scientific Computing.

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

# Appendix

We now present detailed theoretical background, proofs, experiment settings, and additional results that complement the main paper. Appendix A reviews essential background on the infinite-width limit of neural networks [12] and how they relate to Gaussian processes [13]. We discuss the distinction and applicability of the two main kernel-based frameworks suitable for this type of analysis, namely, the neural tangent kernel (NTK) by Jacot et al. [11] and the conjugate kernel, also known as the neural network Gaussian process (NNGP) kernel [38, 40], which was the main tool for our derivations. Appendix B formulates Fourier neural operators [28] under the mathematical abstraction that allowed us to derive the operator-valued kernel for neural operators. The proofs of the main theoretical results then appear in Appendix C, including the construction and properties of the operator-valued kernel and the correspondence between trained neural operators and their GP limits. Appendix D describes the PDE benchmarks considered, namely Darcy flow and shallow water equations, alongside the respective objective functionals for optimization tasks. Experiment details, hyperparameter settings, and baseline implementation details are provided in Section D.4. Appendix E presents results on an experiment with a single-hidden-layer neural operator validating our theoretical results. Lastly, we discuss limitations and potential broader impact in sections F and G, respectively.

## A    Additional background

In this section, we discuss the main differences between the neural tangent kernel [11] and the conjugate kernel, also known as the neural network Gaussian process (NNGP) kernel [12]. Both kernels are used to approximate the behavior of neural networks, but they differ in how they use Gaussian processes to describe the network's behavior.

### A.1    Conjugate kernel (NNGP)

The conjugate kernel has long been studied in the neural networks literature, describing the correspondence neural networks with randomized parameters and their limiting distribution as the network width approaches infinity [38–41, 54]. Neal [40] first showed the correspondence between an infinitely wide single-hidden-layer network and a Gaussian process by applying the central limit theorem. More recent works [38, 41, 54] later showed that the same reasoning can be extended to neural networks with multiple hidden layers. The NNGP kernel is particularly useful for Bayesian inference as it allows us to define GP priors for neural networks and analyze how they change when conditioned on data, providing us with closed-form expressions for an exact GP posterior in the infinite-width limit [38].

Define an $L$-layer neural network $h(\cdot, \boldsymbol{\theta}) : \mathcal{X} \to \mathbb{R}$ with $h(x; \boldsymbol{\theta}) := h_L(x; \boldsymbol{\theta})$ via the recursion:

$$
\begin{aligned}
h_0(x; \boldsymbol{\theta}) &:= x \\
h_l(x; \boldsymbol{\theta}) &:= \alpha_l(\mathbf{W}_l h_{l-1}(x; \boldsymbol{\theta}) + \mathbf{b}_l), \quad l \in \{1, \dots, L\},
\end{aligned}
\tag{16}
$$

where $x \in \mathcal{X}$ represents an arbitrary input on a finite-dimensional domain $\mathcal{X}$, $\mathbf{W}_l \in \mathbb{R}^{M_l \times M_{l-1}}$ denotes a layer's weights matrix, $M_l$ is the width of the $l$th layer, $\mathbf{b}_l \in \mathbb{R}^{M_l}$ is a bias vector, $\alpha_l : \mathbb{R} \to \mathbb{R}$ denotes the layer's activation function, which is applied elementwise on vector-valued inputs, and $\boldsymbol{\theta} := \mathrm{vec}(\{\mathbf{W}_l, b_l\}_{l=1}^L)$ collects all the network parameters into a vector. Assume $[\mathbf{W}_l]_{i,j} \sim \mathcal{N}\left(0, \frac{1}{M_{l-1}}\right)$ and $[\mathbf{b}_l]_i \sim \mathcal{N}(0, 1)$, for $i \in \{1, \dots, M_l\}$, $j \in \{1, \dots, M_{l-1}\}$ and $l \in \{1, \dots, L\}$, and let $M := \min\{M_1, \dots, M_L\}$. The NNGP kernel then corresponds to the infinite-width limit of the network outputs covariance function [38] as:

$$
k_{\mathrm{NNGP}}(x, x') := \lim_{M \to \infty} \mathbb{E}[h(x; \boldsymbol{\theta})h(x'; \boldsymbol{\theta})], \quad x, x' \in \mathcal{X},
\tag{17}
$$

where the expectation is taken under the parameters distribution. By an application of the central limit theorem, it can be shown [38, 40] that the neural network converges in distribution to a Gaussian process with the kernel defined above, i.e.:

$$
h_{\boldsymbol{\theta}} \xrightarrow{d} h \sim \mathcal{GP}(0, k_{\mathrm{NNGP}}),
\tag{18}
$$

where $\xrightarrow{d}$ denotes convergence in distribution as $M \to \infty$. In other words, the randomly initialized network follows a GP prior in the infinite-width limit. Moreover, it follows that, when conditioned on

data $\mathcal{D}_N := \{x_i, y_i\}_{i=1}^N$, assuming $y_i = h(x_i) + \epsilon_i$ and $\epsilon_i \sim \mathcal{N}(0, \sigma_\epsilon^2)$, a Bayesian neural network is distributed according to a GP posterior in the infinite-width limit as:

$$h|\mathcal{D}_N \sim \mathcal{GP}(\mu_N, k_N) \tag{19}$$

$$\mu_N(x) := \mathbb{E}[h(x) \mid \mathcal{D}_N] = \mathbf{k}_N(x)^\mathsf{T}(\mathbf{K}_N + \sigma_\epsilon^2 \mathbf{I})^{-1}\mathbf{y}_N \tag{20}$$

$$k_N(x, x') := \mathrm{Cov}[h(x), h(x') \mid \mathcal{D}_N] = k(x, x') - \mathbf{k}_N(x)^\mathsf{T}(\mathbf{K}_N + \sigma_\epsilon^2 \mathbf{I})^{-1}\mathbf{k}_N(x'), \tag{21}$$

for any $x, x' \in \mathcal{X}$, where $\mathbf{K}_N := [k(x_i, x_j)]_{i,j=1}^N \in \mathbb{R}^{N \times N}$, $\mathbf{k}_N(x) := [k(x_i, x)]_{i=1}^N \in \mathbb{R}^N$, $\mathbf{y}_N := [y_i]_{i=1}^N$, and we set $k := k_{\texttt{NNGP}}$ to avoid notation clutter. Hence, the NNGP kernel allows us to compute exact GP posteriors for neural network models. However, we emphasize that the conjugate kernel should not be confused with the neural tangent kernel [11], which corresponds to the infinite-width limit of $\mathbb{E}[\nabla_{\boldsymbol{\theta}} h(x; \boldsymbol{\theta}) \cdot \nabla_{\boldsymbol{\theta}} h(x'; \boldsymbol{\theta})]$, instead.

## A.2  Neural tangent kernel (NTK)

The NTK approximates the behavior of a neural network during training via gradient descent by considering the gradients of the network with respect to its parameters [11]. Consider an $L$-layer feedforward neural network $h_{\boldsymbol{\theta}} : \mathcal{X} \to \mathbb{R}$ as defined in Eq. 16. In its original formulation, Jacot et al. [11] applied a scaling factor of $\frac{1}{\sqrt{M}}$ to the output of each layer to ensure asymptotic convergence in the limit $M \to \infty$ of the network trained via gradient descent. However, later works showed that standard network parameterizations (without explicit output scaling) also converge to the same limit as long as a LeCun or Kaiming/He type of initialization scheme is applied to the parameters with appropriate scaling of the learning rates [12, 55], which ensure bounded variance in the infinite-width limit. The NTK describes the limit:

$$k_{\texttt{NTK}}(x, x') = \lim_{M \to \infty} \mathbb{E}[\nabla_{\boldsymbol{\theta}} h_{\boldsymbol{\theta}}(x) \cdot \nabla_{\boldsymbol{\theta}} h_{\boldsymbol{\theta}}(x')], \tag{22}$$

for any $x, x' \in \mathcal{X}$, where the expectation is taken under the parameters initialization distribution. Under mild assumptions, the trained network's output distribution converges to a Gaussian process described by the NTK [11, 38]. Although originally derived for the unregularized case, applying L2 regularization to the parameters norm yields a GP posterior with a term that can account for observation noise [35]. Namely, consider the following loss function:

$$\ell_N(\boldsymbol{\theta}) := \sum_{i=1}^N (y_i - h_{\boldsymbol{\theta}}(x_i))_2^2 + \lambda \|\boldsymbol{\theta} - \boldsymbol{\theta}_0\|_2^2, \tag{23}$$

where $\boldsymbol{\theta}_0$ denotes the initial parameters. As the network width grows larger, the NTK tells us that the network behaves like a linear model [11, 55] as:

$$h(x; \boldsymbol{\theta}) \approx h(x; \boldsymbol{\theta}_0) + \nabla_{\boldsymbol{\theta}} h(x; \boldsymbol{\theta})\big|_{\boldsymbol{\theta} := \boldsymbol{\theta}_0} \cdot (\boldsymbol{\theta} - \boldsymbol{\theta}_0), \quad x \in \mathcal{X}. \tag{24}$$

The approximation becomes exact in the infinite width limit within any bounded neighborhood $\mathcal{B}_R(\boldsymbol{\theta}_0) := \{\boldsymbol{\theta} \mid \|\boldsymbol{\theta} - \boldsymbol{\theta}_0\| \leq R\}$ of arbitrary radius $0 < R < \infty$ around $\boldsymbol{\theta}_0$, as the second-order error term vanishes [55]. The latter also means that $\nabla_{\boldsymbol{\theta}} h(\cdot; \boldsymbol{\theta})$ converges to fixed feature map $\phi : \mathcal{X} \to \mathcal{H}_0$, where $\mathcal{H}_0$ is the Hilbert space spanned by the limiting gradient vectors. With this observation, our loss function can be rewritten as:

$$\ell_N(\boldsymbol{\theta}) \approx \sum_{i=1}^N \left(y_i - h(x_i; \boldsymbol{\theta}_0) - \nabla_{\boldsymbol{\theta}} h(x_i; \boldsymbol{\theta})\big|_{\boldsymbol{\theta} := \boldsymbol{\theta}_0} \cdot (\boldsymbol{\theta} - \boldsymbol{\theta}_0)\right)^2 + \lambda \|\boldsymbol{\theta} - \boldsymbol{\theta}_0\|_2^2. \tag{25}$$

The minimizer of the approximate loss can be derived in closed form. Applying the NTK then yields the infinite-width model:

$$h_N(x) = h(x) + \mathbf{k}_N^{\texttt{NTK}}(x)^\mathsf{T}(\mathbf{K}_N^{\texttt{NTK}} + \lambda \mathbf{I})^{-1}(\mathbf{y}_N - \mathbf{h}_N), \tag{26}$$

where $h \sim \mathcal{GP}(0, k_{\texttt{NNGP}})$ denotes the network at its random initialization, as defined above, $\mathbf{k}_N^{\texttt{NTK}}(x) := [k_{\texttt{NTK}}(x_i, x)]_{i=1}^N \in \mathbb{R}^N$, $\mathbf{K}_N^{\texttt{NTK}} := [k_{\texttt{NTK}}(x_i, x_j)]_{i,j=1}^N \in \mathbb{R}^{N \times N}$, and $\mathbf{h}_N := [h(x_i)]_{i=1}^N \in \mathbb{R}^N$. Now applying the GP limit to the randomly initialized network $h$ [12, 35], we have that:

$$h_N \sim \mathcal{GP}(\hat{\mu}_N, \hat{k}_N) \tag{27}$$

$$\hat{\mu}_N(x) = \mathbf{k}_N^{\texttt{NTK}}(x)^\mathsf{T}(\mathbf{K}_N^{\texttt{NTK}} + \lambda \mathbf{I})^{-1}\mathbf{y}_N \tag{28}$$

$$\hat{k}_N(x, x') = k(x, x') + \mathbf{k}_N^{\texttt{NTK}}(x)^\mathsf{T}(\mathbf{K}_N^{\texttt{NTK}} + \lambda \mathbf{I})^{-1}\mathbf{K}_N(\mathbf{K}_N^{\texttt{NTK}} + \lambda \mathbf{I})^{-1}\mathbf{k}_N^{\texttt{NTK}}(x')$$
$$- \mathbf{k}_N^{\texttt{NTK}}(x)^\mathsf{T}(\mathbf{K}_N^{\texttt{NTK}} + \lambda \mathbf{I})^{-1}\mathbf{k}_N(x') - \mathbf{k}_N(x)^\mathsf{T}(\mathbf{K}_N^{\texttt{NTK}} + \lambda \mathbf{I})^{-1}\mathbf{k}_N^{\texttt{NTK}}(x'), \tag{29}$$

where we again set $k := k_{\text{NNGP}}$ to avoid clutter. However, note that such GP model does not generally correspond to a Bayesian posterior. An exception is where only the last linear layer is trained, while the rest are kept fixed at their random initialization; in which case case, the GP described by the NTK and the exact GP posterior according to the NNGP kernel match in the unregularized setting [12].

### A.3 Application to Thompson sampling

For our purpose, it is important to have a Bayesian posterior in order to apply Gaussian process Thompson sampling (GP-TS) [33] for the regret bounds in Proposition 2. Therefore, we are constrained by existing theories connecting neural networks to Gaussian processes to assume training only the last layer of neural networks of infinite width, which gives a Bayesian posterior of the NNGP after training. In addition, we had to consider the case of a single hidden layer neural operator, as the usual recursive step applied to derive the infinite-width limit would require an intermediate (infinite-dimensional) function space in our case, making the extension to the multi-layer case not trivial due to the usual finite-dimensional assumptions [55]. Nonetheless, the NOTS algorithm suggested by our theory has demonstrated competitive performance in our experiments even in more relaxed settings with a multi-layer model. Future theoretical developments in Bayesian analysis of neural networks may eventually permit the convergence analysis of the more relaxed settings in our experiments. In any case, we present an experiment with a wide single-hidden-layer model with training only on the last layer in Appendix E.

## B    Fourier neural operators under the abstract representation

Recalling the definition in the main paper, we consider a single hidden layer neural operator. Let $M \in \mathbb{N}$ represent the layer's width, $A_{\mathbf{R}} : \mathcal{A} \to \mathcal{C}(\mathcal{Z}, \mathbb{R}^{d_{\mathbf{R}}})$ denote a (fixed) continuous operator, and $b_0 : \mathcal{Z} \to \mathbb{R}^{d_b}$ denote a (fixed) continuous function. For simplicity, we assume scalar outputs with $d_u = 1$. We consider models of the form:

$$G_{\boldsymbol{\theta}}(a)(z) = \mathbf{w}_o^{\mathsf{T}} \alpha \left( \mathbf{W}_{\mathbf{R}} A_{\mathbf{R}}(a)(z) + \mathbf{W}_u a(\Pi_0(z)) + \mathbf{W}_b b_0(z) \right), \quad z \in \mathcal{Z}, \qquad (30)$$

where $\boldsymbol{\theta} := (\mathbf{w}_o, \mathbf{W}_{\mathbf{R}}, \mathbf{W}_u, \mathbf{W}_b) \in \mathbb{R}^M \times \mathbb{R}^{M \times d_{\mathbf{R}}} \times \mathbb{R}^{M \times d_a} \times \mathbb{R}^{M \times d_b} =: \mathcal{W}$ represents parameters.

**Fourier neural operators.**    As an example, we show how the formulation above applies to the Fourier neural operator (FNO) architecture [28]. For simplicity, assume that $\mathcal{X}$ is the $d$-dimensional periodic torus, i.e., $\mathcal{X} = [0, 2\pi)^d$, and $\mathcal{Z} = \mathcal{X}$. Then any square-integrable function $a : \mathcal{X} \to \mathbb{C}^{d_a}$ can be expressed as a Fourier series:

$$a(x) = \sum_{s \in \mathbb{Z}^d} \hat{a}(s) e^{\iota \langle s, x \rangle}, \quad \forall x \in \mathcal{X}, \qquad (31)$$

where $\iota := \sqrt{-1} \in \mathbb{C}$ denotes the imaginary unit, and $\hat{a}(s)$ are coefficients given by the function's Fourier transform $F : \mathcal{L}^2(\mathcal{X}, \mathbb{C}^{d_a}) \to \mathcal{L}^2(\mathbb{Z}^d, \mathbb{C}^{d_a})$ as:

$$\hat{a}(s) := (Fa)(s) = \frac{1}{(2\pi)^d} \int_{\mathcal{X}} a(x) e^{-\iota \langle s, x \rangle} \, \mathrm{d}x, \quad s \in \mathbb{Z}^d. \qquad (32)$$

For a translation-invariant kernel $\mathbf{R}(x, x') = \mathbf{R}(x - x')$, applying the convolution theorem, the integral operator can be expressed as:

$$\int_{\mathcal{X}} \mathbf{R}(\cdot, x) a(x) \, \mathrm{d}x = \mathbf{R} * a$$
$$= F^{-1}(F(\mathbf{R}) \cdot F(a)) \qquad (33)$$
$$= \sum_{s \in \mathbb{Z}^d} \widehat{\mathbf{R}}(s) \hat{a}(s) e^{\iota \langle s, \cdot \rangle}$$

In practice, function observations are only available at a discrete set of points and the Fourier series is truncated at a maximum frequency $s_{\max} \in \mathbb{Z}^d$, which allows one to efficiently compute it via the fast Fourier transform (FFT). Considering these facts, FNOs approximate the integral as [28]:

$$\int_{\mathcal{X}} \mathbf{R}(x, x') a(x') \, \mathrm{d}x' \approx \sum_{n=1}^{N} \widehat{\mathbf{R}}(s_n) \hat{a}(s_n) e^{\iota \langle s_n, x \rangle}, \quad x \in \mathcal{Z}, \qquad (34)$$

where the $N$ values of $s_n$ range from 0 to $s_{\max}$ in all $d$ coordinates. Finally, defining $A_{\mathbf{R}}$ as:

$$A_{\mathbf{R}} : \mathcal{C}(\mathcal{X}, \mathbb{C}^{d_a}) \to \mathcal{C}(\mathcal{X}, \mathbb{C}^{Nd_a})$$

$$a \mapsto \begin{bmatrix} (Fa)(s_1)e^{\iota\langle s_1, \cdot \rangle} \\ \vdots \\ (Fa)(s_N)e^{\iota\langle s_N, \cdot \rangle} \end{bmatrix}, \tag{35}$$

and letting $\mathbf{W_R} = [\widehat{\mathbf{R}}(s_1), \dots, \widehat{\mathbf{R}}(s_N)]$, we recover Eq. 30 for FNOs in the complex-valued case.

For real-valued functions, to ensure that the result is again real-valued, a symmetry condition is imposed on $\widehat{\mathbf{R}}$, so that its values for negative frequencies are the conjugate transpose of the corresponding values for positive frequencies. However, we can still represent it via a single matrix of weights, which is simply conjugate transposed for the negative frequencies. Lastly, note that complex numbers can be represented as tuples of real numbers.

## C  Theoretical Analysis

In this section, we provide the proofs of the theoretical results presented in the main paper.

### C.1  Auxiliary results

**Definition 1** (Multi-Layer Fully-Connected Neural Network). *A multi-layer fully-connected neural network with L hidden layers, input dimension $d_0$, output dimension $d_{L+1}$, and hidden layer widths $d_1, \dots, d_L$, is defined recursively as follows. For input $x \in \mathcal{X}$, the pre-activations and activations at layer $l = 1, \dots, L+1$ are:*

$$\mathbf{v}^{(1)}(x) = \mathbf{W}^{(0)} x + \mathbf{b}^{(0)} \tag{36}$$

$$\mathbf{v}^{(l)}(x) = \mathbf{W}^{(l-1)} \alpha(\mathbf{v}^{(l-1)}(x)) + \mathbf{b}^{(l-1)}, \quad l = 2, \dots, L, \tag{37}$$

$$\mathbf{v}^{(L+1)}(x) = \mathbf{W}^{(L)} \alpha(\mathbf{v}^{(L)}(x)), \tag{38}$$

*where $\mathbf{W}^{(l)} \in \mathbb{R}^{d_{l+1} \times d_l}$ are weight matrices, $\mathbf{b}^{(l)} \in \mathbb{R}^{d_{l+1}}$ are bias vectors, $\alpha : \mathbb{R} \to \mathbb{R}$ is a coordinate-wise non-linearity, and the network output is $f(x) = \mathbf{v}^{(L+1)}(x)$. The weights are initialized as $W_{ij}^{(l)} = \left(\frac{c_W}{d_l}\right)^{1/2} \widehat{W}_{ij}^{(l)}$, where $\widehat{W}_{ij}^{(l)} \sim \mu$ with mean 0, variance 1, and finite higher moments, and biases as $b_i^{(l)} \sim \mathcal{N}(0, c_b)$, given fixed constants $c_W > 0$ and $c_b \geq 0$.*

**Lemma 1** (Infinite-width limit [56]). *Consider a feedforward fully connected neural network as in Definition 1 with non-linearity $\alpha : \mathbb{R} \to \mathbb{R}$ that is absolutely continuous with polynomially bounded derivative. Fix the input dimension $d_0$, the output dimension $d_{L+1}$, the number of layers $L$, and a compact set $\mathcal{X} \subset \mathbb{R}^{d_0}$. As hidden layer widths $d_1, \dots, d_L \to \infty$, the random field $x \mapsto f(x)$ converges weakly in $\mathcal{C}(\mathcal{X}, \mathbb{R}^{d_{L+1}})$ to a centered Gaussian process with covariance $\mathbf{K}^{(L+1)} : \mathcal{X} \times \mathcal{X} \to \mathbb{R}^{d_{L+1} \times d_{L+1}}$ defined recursively by:*

$$\mathbf{K}^{(l+1)}(x, x') = c_b \mathbf{I} + c_W \, \mathbb{E}_{(\mathbf{v}, \mathbf{v}')} \left[ \alpha(\mathbf{v}) \otimes \alpha(\mathbf{v}') \right], \tag{39}$$

*where $(\mathbf{v}, \mathbf{v}') \sim \mathcal{N}\left(0, \begin{bmatrix} \mathbf{K}^{(l)}(x, x) & \mathbf{K}^{(l)}(x, x') \\ \mathbf{K}^{(l)}(x, x') & \mathbf{K}^{(l)}(x', x') \end{bmatrix}\right)$ for $l \geq 2$, with the initial condition for $l = 1$ determined by the first-layer weights and biases.*

**Lemma 2** (Thm. 3.1 in Takeno et al. [33]). *Let $f \sim \mathcal{GP}(0, k)$, where $k : \mathcal{X} \times \mathcal{X} \to \mathbb{R}$ is a positive-definite kernel on a finite $\mathcal{X}$. Then the Bayesian cumulative regret of GP-TS is such that:*

$$R_T \in \mathcal{O}(\sqrt{T \gamma_T}),$$

*where $\gamma_T$ denotes the maximum information gain after $T$ iterations with the GP model.*

### C.2  Infinite-width neural operator kernel

**Assumption 1.** *The activation function $\alpha : \mathbb{R} \to \mathbb{R}$ is absolutely continuous with derivative bounded almost everywhere.*

**Lemma 3** (Continuity of limiting GP). *Let $G_{\boldsymbol{\theta}} : \mathcal{A} \to \mathcal{C}(\mathcal{Z})$ be a neural operator with a single hidden layer, as defined as in Eq. 30. Assume $\mathbf{w}_o \sim \mathcal{N}(\mathbf{0}, \sigma_{\boldsymbol{\theta}}^2 \mathbf{I})$, for $\sigma_{\boldsymbol{\theta}}^2 > 0$ such that $\sigma_{\boldsymbol{\theta}}^2 \propto \frac{1}{M}$, and let the remaining parameters have their entries sampled from a fixed normal distribution. Then, as $M \to \infty$, the neural operator converges in distribution to a zero-mean Gaussian process with continuous realizations $G : \mathcal{A}' \to \mathcal{C}(\mathcal{Z})$ on every compact subset $\mathcal{A}' \subset \mathcal{A}$.*

*Proof.* As shown in Section 5.2, when evaluated at a fixed point $z \in \mathcal{Z}$, a neural operator with a single hidden layer can be seen as:

$$G_{\boldsymbol{\theta}}(a)(z) = h_{\boldsymbol{\theta}}(\boldsymbol{\psi}(a, z)), \quad a \in \mathcal{A}, \tag{40}$$

where $\boldsymbol{\psi}(a, z) := \mathbf{v}_z(a)$ is a fixed map $\boldsymbol{\psi} : \mathcal{A} \times \mathcal{Z} \to \mathcal{V}$, with $\mathcal{V} = \mathbb{R}^{d_\mathbf{R} + d_a + d_b}$, and $h_{\boldsymbol{\theta}}$ is a conventional feedforward neural network, as defined in Definition 1. By Assumption 1 and Lemma 1, it follows that, as $M \to \infty$, $h_{\boldsymbol{\theta}}$ converges in distribution to a Gaussian process $h \sim \mathcal{GP}(0, k_h)$ with continuous sample paths, i.e., $\mathbb{P}[h \in \mathcal{C}(\mathcal{V}')] = 1$ on every compact $\mathcal{V}' \subset \mathcal{V}$. The continuity of $\boldsymbol{\psi} : \mathcal{A} \times \mathcal{Z} \to \mathcal{V}$ then implies that $g := h \circ \boldsymbol{\psi}$ is a zero-mean GP whose sample paths lie almost surely in $\mathcal{C}(\mathcal{A}' \times \mathcal{Z})$, for a compact $\mathcal{A}' \subset \mathcal{A}$, as $\mathcal{Z}$ is already assumed compact. Therefore, for each $a \in \mathcal{A}$, we have $\mathbb{P}[g(a, \cdot) \in \mathcal{C}(\mathcal{Z})] = 1$, so that $G(a) := g(a, \cdot)$ defines an almost surely continuous operator $G : \mathcal{A}' \to \mathcal{C}(\mathcal{Z})$ on compact $\mathcal{A}' \subset \mathcal{A}$. The verification that $G$ is a vector-valued GP trivially follows. $\qquad\square$

**Proposition 1.** *Let $G_{\boldsymbol{\theta}} : \mathcal{A} \to \mathcal{U}$ be a neural operator with a single hidden layer, where $\mathcal{U} \subseteq \mathcal{L}^2(\nu)$ is closed, and $\nu$ is a finite Borel measure on $\mathcal{Z}$. Assume $\mathbf{w}_o \sim \mathcal{N}(\mathbf{0}, \sigma_{\boldsymbol{\theta}}^2 \mathbf{I})$, for $\sigma_{\boldsymbol{\theta}}^2 > 0$ such that $\sigma_{\boldsymbol{\theta}}^2 \propto 1/M$, while the remaining parameters have their entries sampled from a fixed normal distribution. Then, as $M \to \infty$, on every compact subset of $\mathcal{A}$, the neural operator converges in distribution to a zero-mean vector-valued Gaussian process with operator-valued covariance function given by:*

$$\lim_{M \to \infty} \mathbb{E}_{\boldsymbol{\theta} \sim \mathcal{N}(\mathbf{0}, \boldsymbol{\Sigma}_0)}[G_{\boldsymbol{\theta}}(a) \otimes G_{\boldsymbol{\theta}}(a')] = K_G(a, a'), \quad a, a' \in \mathcal{A},$$

*where $K_G : \mathcal{A} \times \mathcal{A} \to \mathcal{L}(\mathcal{U})$ is defined in Eq. 12, and $\otimes$ denotes the outer product.*

*Proof of Proposition 1.* We start by noting that any continuous function $u \in \mathcal{C}(\mathcal{Z})$ is automatically included in $\mathcal{L}^2(\nu)$, since $\|u\|_{\mathcal{L}^2(\nu)}^2 = \int_{\mathcal{Z}} u^2(z) \, \mathrm{d}\nu(z) \leq \nu(\mathcal{Z}) \|u\|_\infty^2 < \infty$. Hence, any operator mapping into $\mathcal{C}(\mathcal{Z})$ also maps into $\mathcal{L}^2(\nu)$ by inclusion.

Applying Lemma 3, it follows that $G_{\boldsymbol{\theta}} \xrightarrow{d} G$, where $G$ is a zero-mean GP, as $M \to \infty$. Now, given any $u \in \mathcal{U}$, $a, a' \in \mathcal{A}$ and $z \in \mathcal{Z}$, we have that:

$$
\begin{aligned}
(\mathbb{E}[G(a) \otimes G(a')]u)(z) &= \mathbb{E}[G(a)\langle G(a'), u\rangle] \\
&= \left( \mathbb{E}\left[ g(a, \cdot) \int_{\mathcal{Z}} g(a', z')u(z') \, \mathrm{d}\nu(z') \right] \right)(z) \\
&= \mathbb{E}\left[ \int_{\mathcal{Z}} g(a, z)g(a', z')u(z') \, \mathrm{d}\nu(z') \right] \\
&= \int_{\mathcal{Z}} \mathbb{E}[g(a, z)g(a', z')]u(z') \, \mathrm{d}\nu(z') \\
&= \int_{\mathcal{Z}} k_G(a, z, a', z')u(z') \, \mathrm{d}\nu(z'),
\end{aligned}
\tag{41}
$$

where we applied the linearity of expectations and the correspondence between $g : \mathcal{A} \times \mathcal{Z} \to \mathbb{R}$ and the limiting operator $G : \mathcal{A} \to \mathcal{U}$. As the choice of elements was arbitrary, it follows that the above defines an operator-valued kernel $K_G$. Linearity follows from the expectations. Given any $a \in \mathcal{A}$, as a positive-semidefinite operator, the operator norm of $K_G(a, a)$ is bounded by its trace, such that:

$$\|K_G(a, a)\| \leq \mathrm{Tr}(K_G(a, a)) = \mathbb{E}[\|G(a)\|_{\mathcal{U}}^2] = \mathbb{E}\left[ \int_{\mathcal{Z}} g^2(a, z) \, \mathrm{d}\nu(z) \right] < \nu(\mathcal{Z}) \mathbb{E}[\|g(a, \cdot)\|_\infty^2], \tag{42}$$

and the last expectation is finite, since $g$ is almost surely continuous. Hence, $K_G(a, a) \in \mathcal{L}(\mathcal{U})$. $\qquad\square$

## C.3 Regret bound

**Proposition 2.** *Let $f : \mathcal{U} \to \mathbb{R}$ be a bounded linear functional such that $f = \tilde{f} \circ H$, where $\tilde{f} : \mathcal{Y} \to \mathbb{R}$ is linear, and $G_* \sim \mathcal{GP}(0, K)$. Consider a sequential algorithm selecting $a_t \in \arg\max_{a \in \mathcal{S}} f(G_t(a))$ and observing $y_t = HG_*(a_t) + \xi_t$, where $G_t \overset{d}{=} G_* | \mathcal{D}_t$, and $\xi_t \sim \mathcal{N}(0, \lambda I)$, for $t \in \{1, \ldots, T\}$. Then, this algorithm's expected cumulative regret is such that:*

$$R_T \in \tilde{\mathcal{O}}(\sqrt{T}), \tag{15}$$

*where $\tilde{\mathcal{O}}(\cdot)$ suppresses logarithmic factors of the $\mathcal{O}(\cdot)$ asymptotic rate.*

*Proof of Proposition 2.* Starting with the assumption that $f = \tilde{f} \circ H$, an observation $\mathbf{y} = HG_*(a) + \boldsymbol{\xi}$ only provides information about $HG_* : \mathcal{A} \to \mathcal{Y}$, missing any component of $G_*$ mapping to the null space $\ker(H) \subset \mathcal{U}$ of the observation operator $H$. Thus, any $\tilde{G} = G_* + Z$ is indistinguishable from $G_*$, for any $Z : \mathcal{A} \to \mathcal{U}$ with range $Z(\mathcal{A}) \subset \ker(H)$, based on the information available in the observations. Therefore, for the optimization objective $f \circ G_* : \mathcal{A} \to \mathbb{R}$ to be identifiable, we restrict admissible functionals such that $f(u + \omega) = f(u)$, for all $\omega \in \ker(H)$ and $u \in \mathcal{U}$, and assuming $f = \tilde{f} \circ H$ ensures that this requirement is satisfied.

By linearity, it follows that $f \circ G_* \sim \mathcal{GP}(0, k_f)$ for a fixed bounded linear functional $f : \mathcal{U} \to \mathbb{R}$. Hence, $f \circ G_*$ is equal in distribution to a scalar-valued GP $h \sim \mathcal{GP}(0, k_f)$ with $k_f : \mathcal{A} \times \mathcal{A} \to \mathbb{R}$ given by:

$$k_f(a, a') = f(K(a, a')f), \quad a, a' \in \mathcal{A},$$

where we implicitly identify the functional $f$ with a unique corresponding vector in $\mathcal{U}$, also denoted by $f$, by the Riesz representation theorem to apply the operator $K(a, a') \in \mathcal{L}(\mathcal{U})$ to $f$. By Lemma 2, standard GP-TS on an objective $h \sim \mathcal{GP}(0, k_f)$, a finite domain $\mathcal{S} \subset \mathcal{A}$ will have Bayesian cumulative regret $R_T \in \mathcal{O}(\sqrt{T\gamma_{f,T}})$. Note that $\gamma_{f,T}$, in our case, corresponds to the maximum information gain after $T$ vector-valued observations $y_t \in \mathcal{Y} \subseteq \mathbb{R}^m$, for $t \in \{1, \ldots, T\}$, not a scalar as it would be usually assumed in GP-TS. However, the proof of Lemma 2 in Takeno et al. [33, Thm. 3.1] does not depend on the particular form of the posterior mean $\mathbb{E}[h(a) \mid \mathcal{D}_t]$ or variance $\mathbb{V}[h(a) \mid \mathcal{D}_t]$, as long as the posterior remains a GP, which still holds. Lastly, we analyze the information gain.

The information gain about $h = \tilde{f} \circ HG_*$ after $T$ observations is given by the mutual information $\mathbb{I}(y_{1:T}; h)$. As $h$ is a deterministic function of $G_*$, by the data-processing inequality [see 57, Thm. 2.8.1],

$$\mathbb{I}(y_{1:T}; h) \leq \mathbb{I}(y_{1:T}; G_*).$$

In addition, there are at most $|\mathcal{S}| < \infty$ distinct (in distribution) random variables $y_a$, for $a \in \mathcal{S}$, which allows us to provide a generic upper bound on the growth rate of the information gain.

Mutual information is invariant under permutations. Then, for large enough $T$, we can rearrange the observations as:

$$\mathbb{I}(y_{1:T}; G_*) = \mathbb{I}(\{y_a^1, \ldots, y_a^{n_{T,a}}\}_{a \in \mathcal{S}}; G_*),$$

where $y_a^i$ denotes the $i$'th observation with input function $a \in \mathcal{S}$, for $i \in \{1, \ldots, n_{T,a}\}$, and $\sum_{a \in \mathcal{S}} n_{T,a} = T$. In addition, observation $y_a$ is independent of observation $y_{a'}$, for $a \neq a' \in \mathcal{S}$, when conditioned on $G_*$ (and the respective inputs $a, a'$, which we are omitting to avoid notation clutter). By the chain rule of mutual information, we then have that:

$$\mathbb{I}(\{y_a^1, \ldots, y_a^{n_{T,a}}\}_{a \in \mathcal{S}}; G_*) = \sum_{a \in \mathcal{S}} \mathbb{I}(y_a^1, \ldots, y_a^{n_{T,a}}; G_*).$$

The summand corresponds to the information gain after $n_{T,a}$ repeated observations at the same $a$,

$$\mathbb{I}(y_a^1, \ldots, y_a^{n_{T,a}}; G_*) = \frac{1}{2} \log \det(\mathbf{I} + \lambda^{-1}\mathbf{K}_{T,a}),$$

where $\mathbf{K}_{T,a} = [HK(a, a)H^\mathsf{T}]_{i,j=1}^{n_{T,a}} \in \mathbb{R}^{mn_{T,a} \times mn_{T,a}}$. Thus, $\mathbf{K}_{T,a}$ can have at most only $m$ distinct eigenvalues with multiplicity up to $n_{T,a}$, and the maximum eigenvalue is bounded by the trace $\mathrm{Tr}(HK(a, a)H^\mathsf{T}) \in \mathcal{O}(m)$. Therefore,

$$\log \det(\mathbf{I} + \lambda^{-1}\mathbf{K}_{T,a}) \in \mathcal{O}(m \log(mn_{T,a})) = \mathcal{O}(m \log n_{T,a})$$

Combining all the equations above, we get:

$$\mathbb{I}(y_{1:T}; h) \leq \sum_{a \in \mathcal{S}} \mathbb{I}(y_a^1, \ldots, y_a^{n_{T,a}}; G_*) \in \mathcal{O}\left(\sum_{a \in \mathcal{S}} m \log(n_{T,a})\right) = \mathcal{O}(|\mathcal{S}| m \log(T)),$$

as $n_{T,a} \leq T$. Hence, $\gamma_{f,T}$ is $\mathcal{O}(\log T)$, given that $m$ and $|\mathcal{S}|$ are fixed. As a result the cumulative regret is:

$$R_T \in \mathcal{O}(\sqrt{T \gamma_{f,T}}) = \mathcal{O}(\sqrt{m|\mathcal{S}|T \log T}) = \tilde{\mathcal{O}}(\sqrt{T}),$$

with $\tilde{\mathcal{O}}$ suppressing logarithmic factors, which concludes the proof. $\square$

**Remark 1.** *Despite the result above assuming that $f$ is only a function of $G(a)$, there is a straight-forward extension to functionals of the form $f : \mathcal{U} \times \mathcal{A} \to \mathbb{R}$, as considered in our experiments. We simply need to replace $G : \mathcal{A} \to \mathcal{U}$ with the operator $G' : a \mapsto (G(a), a)$ by a concatenation with the identity map $a \mapsto a$, which is deterministic. A similar result then is possible with minor adjustments.*

### C.4 Approximate posterior sampling via gradient descent

We briefly review the equivalence between posterior sampling and gradient descent when training only the last (or readout) layer of a neural network under a (regularized) least-squares loss and LeCun (or Kaiming He) initialization in the presence of observation noise. We will mainly combine major results from the NTK and NNGP literature [12, 35, 55] into the setting of our paper. When only the last layer is trained, the feature maps of the NTK and the NNGP coincide [12, App. D], so that we can follow an NTK type of analysis of how the loss function relates to the network's parameters, while the distribution of the trained network is determined by the NNGP kernel. For simplicity, we focus on the case of a standard, fully connected, scalar-valued neural network, noticing that this analysis is readily extensible to the neural operator case by the techniques we use for our main results.

**Random feature model.** When training only the last layer of a neural network, we have the following model at initialization:

$$h_0(x) = \mathbf{w}_0^\top \phi(x), \tag{43}$$

where we assume $\mathbf{w}_0 \sim \mathcal{N}(\mathbf{0}, \frac{1}{M}\mathbf{I})$ for the initial weights of the readout layer, with $M$ representing the network width, and given $x \in \mathcal{X}$, $\phi(x) \in \mathbb{R}^M$ represents the output of the last hidden layer of the neural network, which consists of a *random feature* map $\phi : \mathcal{X} \to \mathbb{R}^M$ under the initialization scheme. Observe that the NNGP kernel is given by:

$$k_{\text{NNGP}}(x, x') := \lim_{M \to \infty} \mathbb{E}[h_0(x)h_0(x')] = \lim_{M \to \infty} \frac{1}{M}\mathbb{E}[\phi(x)^\top \phi(x')], \tag{44}$$

for any $x, x' \in \mathcal{X}$. Note that this is the same limit we obtain if $\mathbf{w}_0 \sim \mathcal{N}(\mathbf{0}, \mathbf{I})$ and $\phi(x)$ is scaled by $\frac{1}{\sqrt{M}}$, as in the NTK parameterization [11]. Hence, to simplify our derivations, we will adopt the latter in the remainder of this subsection.

**Regularized least-squares estimator.** Given $N$ data points $\mathcal{D}_N := \{x_i, y_i\}_{i=1}^N \subset \mathcal{X} \times \mathbb{R}$, we consider the following regularized least-squares loss:

$$\ell_N(\mathbf{w}) := \frac{1}{2}\sum_{i=1}^N (\mathbf{w}^\top \phi(x_i) - y_i)^2 + \frac{\lambda}{2}\|\mathbf{w} - \mathbf{w}_0\|^2 = \frac{1}{2}\|\mathbf{\Phi}^\top \mathbf{w} - \mathbf{y}\|^2 + \frac{\lambda}{2}\|\mathbf{w} - \mathbf{w}_0\|^2, \tag{45}$$

where $\mathbf{\Phi} := [\phi(x)_1, \ldots, \phi(x)_N] \in \mathbb{R}^{M \times N}$, $\mathbf{y} := [y_1, \ldots, y_N]^\top \in \mathbb{R}^N$, $\mathbf{w}_0 \sim \mathcal{N}(\mathbf{0}, \mathbf{I})$, and $\lambda > 0$ is a regularization factor. We note that, in practice, due to the small initialization variance of order $\frac{1}{M}$, the initial weights $\mathbf{w}_0$ will be elementwise very close to zero, especially for large widths $M$. Therefore, we omit $\mathbf{w}_0$ from the regularizer in Eq. 7, as their practical effect is limited, and a simple L2 regularizer is typically efficiently implemented as a weight decay term in optimization algorithms found within modern deep learning frameworks, such as PyTorch [58].

The loss function in Eq. 45 is convex in $\mathbf{w}$ and therefore admits a unique minimizer $\mathbf{w}_N \in \mathbb{R}^M$, which we can derive in closed form as:

$$\nabla \ell_N(\mathbf{w}) = \mathbf{\Phi}(\mathbf{\Phi}^\top \mathbf{w} - \mathbf{y}) + \lambda(\mathbf{w} - \mathbf{w}_0)$$
$$\nabla \ell_N(\mathbf{w})\big|_{\mathbf{w} = \mathbf{w}_N} = \mathbf{0} \implies (\mathbf{\Phi}\mathbf{\Phi}^\top + \lambda \mathbf{I})\mathbf{w}_N = \mathbf{\Phi}\mathbf{y} + \lambda \mathbf{w}_0. \tag{46}$$

For $\lambda > 0$, the matrix on the left-hand side is positive-definite, and therefore invertible, then:

$$\mathbf{w}_N = (\mathbf{\Phi}\mathbf{\Phi}^\mathsf{T} + \lambda\mathbf{I})^{-1}(\mathbf{\Phi}\mathbf{y} + \lambda\mathbf{w}_0)\,. \tag{47}$$

Suppose $\mathbf{w}_0 \sim \mathcal{N}(\mathbf{0}, \mathbf{I})$. Then $\mathbf{w}_N|\mathbf{y} \sim \mathcal{N}(\widehat{\mathbf{w}}_N, \widehat{\mathbf{\Sigma}}_N)$, where:

$$\widehat{\mathbf{w}}_N := \mathbb{E}[\mathbf{w}_N \mid \mathbf{y}] = (\mathbf{\Phi}\mathbf{\Phi}^\mathsf{T} + \lambda\mathbf{I})^{-1}\mathbf{\Phi}\mathbf{y}\,, \tag{48}$$

and the covariance matrix is given by:

$$
\begin{aligned}
\widehat{\mathbf{\Sigma}}_N := \mathbb{V}[\mathbf{w}_N \mid \mathbf{y}] &= \mathbb{V}[(\mathbf{\Phi}\mathbf{\Phi}^\mathsf{T} + \lambda\mathbf{I})^{-1}(\mathbf{\Phi}\mathbf{y} + \lambda\mathbf{w}_0) \mid \mathbf{y}] \\
&= \mathbb{V}[\lambda(\mathbf{\Phi}\mathbf{\Phi}^\mathsf{T} + \lambda\mathbf{I})^{-1}\mathbf{w}_0] \\
&= \lambda^2(\mathbf{\Phi}\mathbf{\Phi}^\mathsf{T} + \lambda\mathbf{I})^{-1}\mathbb{V}[\mathbf{w}_0](\mathbf{\Phi}\mathbf{\Phi}^\mathsf{T} + \lambda\mathbf{I})^{-1} \\
&= \lambda^2(\mathbf{\Phi}\mathbf{\Phi}^\mathsf{T} + \lambda\mathbf{I})^{-2}\,,
\end{aligned}
\tag{49}
$$

where we used the fact that $\mathbb{V}[\mathbf{A}\mathbf{w}] = \mathbf{A}\mathbb{V}[\mathbf{w}]\mathbf{A}^\mathsf{T}$ for a random vector $\mathbf{w}$, and we also note that $\mathbb{V}[\mathbf{w}_0 \mid \mathbf{y}] = \mathbb{V}[\mathbf{w}_0]$, given that $\mathbf{w}_0$ is sampled independently of $\mathbf{y}$.

**Alternative derivation.** Another way of deriving the expression above is via the joint distribution between $\mathbf{w}_N$ and $\mathbf{y}$. Assume $\mathbf{y} = \mathbf{\Phi}^\mathsf{T}\mathbf{w}_* + \boldsymbol{\epsilon}$, for some $\mathbf{w}_* \sim \mathcal{N}(\mathbf{0}, \mathbf{I})$ and $\boldsymbol{\epsilon} \sim \mathcal{N}(\mathbf{0}, \sigma_\epsilon^2\mathbf{I})$, so that $\mathbf{\Sigma}_\mathbf{y} := \mathbb{V}[\mathbf{y}] = \mathbf{\Phi}\mathbf{\Phi}^\mathsf{T} + \sigma_\epsilon^2\mathbf{I}$. The joint distribution is:

$$
\begin{bmatrix}\mathbf{w}_N \\ \mathbf{y}\end{bmatrix} \sim \mathcal{N}\left(\begin{bmatrix}\mathbf{0} \\ \mathbf{0}\end{bmatrix}, \begin{bmatrix}(\mathbf{\Phi}\mathbf{\Phi}^\mathsf{T} + \lambda\mathbf{I})^{-1}(\mathbf{\Phi}\mathbf{\Sigma}_\mathbf{y}\mathbf{\Phi}^\mathsf{T} + \lambda^2\mathbf{I})(\mathbf{\Phi}\mathbf{\Phi}^\mathsf{T} + \lambda\mathbf{I})^{-1} & (\mathbf{\Phi}\mathbf{\Phi}^\mathsf{T} + \lambda\mathbf{I})^{-1}\mathbf{\Phi}\mathbf{\Sigma}_\mathbf{y} \\ \mathbf{\Sigma}_\mathbf{y}\mathbf{\Phi}^\mathsf{T}(\mathbf{\Phi}\mathbf{\Phi}^\mathsf{T} + \lambda\mathbf{I})^{-1} & \mathbf{\Sigma}_\mathbf{y}\end{bmatrix}\right).
\tag{50}
$$

The covariance of the joint distribution is obtained from the linear relation between $\mathbf{w}_N$ and $\mathbf{y}$ as:

$$
\mathbf{\Sigma}_{\mathbf{w}_N,\mathbf{y}} = \begin{bmatrix}(\mathbf{\Phi}\mathbf{\Phi}^\mathsf{T} + \lambda\mathbf{I})^{-1} & \mathbf{0} \\ \mathbf{0} & \mathbf{I}\end{bmatrix}\left(\begin{bmatrix}\mathbf{\Phi} \\ \mathbf{I}\end{bmatrix}\mathbf{\Sigma}_\mathbf{y}\begin{bmatrix}\mathbf{\Phi} \\ \mathbf{I}\end{bmatrix}^\mathsf{T} + \begin{bmatrix}\lambda^2\mathbf{I} & \mathbf{0} \\ \mathbf{0} & \mathbf{0}\end{bmatrix}\right)\begin{bmatrix}(\mathbf{\Phi}\mathbf{\Phi}^\mathsf{T} + \lambda\mathbf{I})^{-1} & \mathbf{0} \\ \mathbf{0} & \mathbf{I}\end{bmatrix}.
$$

We can see that the matrix above is non-singular and positive definite. In particular, its determinant can be derived as:

$$
\begin{aligned}
\det(\mathbf{\Sigma}_{\mathbf{w}_N,\mathbf{y}}) &= \det\left(\begin{bmatrix}(\mathbf{\Phi}\mathbf{\Phi}^\mathsf{T} + \lambda\mathbf{I})^{-1} & \mathbf{0} \\ \mathbf{0} & \mathbf{I}\end{bmatrix}\right)^2 \det\left(\begin{bmatrix}\mathbf{\Phi} \\ \mathbf{I}\end{bmatrix}\mathbf{\Sigma}_\mathbf{y}\begin{bmatrix}\mathbf{\Phi} \\ \mathbf{I}\end{bmatrix}^\mathsf{T} + \begin{bmatrix}\lambda^2\mathbf{I} & \mathbf{0} \\ \mathbf{0} & \mathbf{0}\end{bmatrix}\right) \\
&= \det(\mathbf{\Phi}\mathbf{\Phi}^\mathsf{T} + \lambda\mathbf{I})^{-2}\det\left(\begin{bmatrix}\mathbf{\Phi}\mathbf{\Sigma}_\mathbf{y}\mathbf{\Phi}^\mathsf{T} & \mathbf{\Phi}\mathbf{\Sigma}_\mathbf{y} \\ \mathbf{\Sigma}_\mathbf{y}\mathbf{\Phi}^\mathsf{T} & \mathbf{\Sigma}_\mathbf{y}\end{bmatrix} + \begin{bmatrix}\lambda^2\mathbf{I} & \mathbf{0} \\ \mathbf{0} & \mathbf{0}\end{bmatrix}\right) \\
&= \det(\mathbf{\Phi}\mathbf{\Phi}^\mathsf{T} + \lambda\mathbf{I})^{-2}\det\left(\begin{bmatrix}\mathbf{\Phi}\mathbf{\Sigma}_\mathbf{y}\mathbf{\Phi}^\mathsf{T} + \lambda^2\mathbf{I} & \mathbf{\Phi}\mathbf{\Sigma}_\mathbf{y} \\ \mathbf{\Sigma}_\mathbf{y}\mathbf{\Phi}^\mathsf{T} & \mathbf{\Sigma}_\mathbf{y}\end{bmatrix}\right) \\
&= \det(\mathbf{\Phi}\mathbf{\Phi}^\mathsf{T} + \lambda\mathbf{I})^{-2}\det(\mathbf{\Sigma}_\mathbf{y})\det(\mathbf{\Phi}\mathbf{\Sigma}_\mathbf{y}\mathbf{\Phi}^\mathsf{T} + \lambda^2\mathbf{I} - \mathbf{\Phi}\mathbf{\Sigma}_\mathbf{y}\mathbf{\Sigma}_\mathbf{y}^{-1}\mathbf{\Sigma}_\mathbf{y}\mathbf{\Phi}^\mathsf{T}) \\
&= \frac{\det(\mathbf{\Sigma}_\mathbf{y})\det(\lambda^2\mathbf{I})}{\det(\mathbf{\Phi}\mathbf{\Phi}^\mathsf{T} + \lambda\mathbf{I})^2} \\
&> 0\,,
\end{aligned}
$$

where the inequality holds as long as $\lambda > 0$ and $\sigma_\epsilon > 0$. Conditioning on $\mathbf{y}$ then yields:

$$\widehat{\mathbf{w}}_N = (\mathbf{\Phi}\mathbf{\Phi}^\mathsf{T} + \lambda\mathbf{I})^{-1}\mathbf{\Phi}\mathbf{y}\,, \tag{51}$$

and:

$$
\begin{aligned}
\widehat{\mathbf{\Sigma}}_N &= (\mathbf{\Phi}\mathbf{\Phi}^\mathsf{T} + \lambda\mathbf{I})^{-1}(\mathbf{\Phi}\mathbf{\Sigma}_\mathbf{y}\mathbf{\Phi}^\mathsf{T} + \lambda^2\mathbf{I})(\mathbf{\Phi}\mathbf{\Phi}^\mathsf{T} + \lambda\mathbf{I})^{-1} - (\mathbf{\Phi}\mathbf{\Phi}^\mathsf{T} + \lambda\mathbf{I})^{-1}\mathbf{\Phi}\mathbf{\Sigma}_\mathbf{y}\mathbf{\Phi}^\mathsf{T}(\mathbf{\Phi}\mathbf{\Phi}^\mathsf{T} + \lambda\mathbf{I})^{-1} \\
&= \lambda^2(\mathbf{\Phi}\mathbf{\Phi}^\mathsf{T} + \lambda\mathbf{I})^{-2}\,.
\end{aligned}
\tag{52}
$$

In contrast, even if $\lambda := \sigma_\epsilon^2$, note that $\widehat{\mathbf{\Sigma}}_N$ does not correspond to the exact posterior covariance, which can be derived as:

$$\begin{bmatrix}\mathbf{w}_* \\ \mathbf{y}\end{bmatrix} \sim \mathcal{N}\left(\begin{bmatrix}\mathbf{0} \\ \mathbf{0}\end{bmatrix}, \begin{bmatrix}\mathbf{I} & \mathbf{\Phi} \\ \mathbf{\Phi}^\mathsf{T} & \mathbf{\Phi}^\mathsf{T}\mathbf{\Phi} + \lambda\mathbf{I}\end{bmatrix}\right). \tag{53}$$

$$\implies \mathbf{\Sigma}_N := \mathbb{V}[\mathbf{w}_* \mid \mathbf{y}] = \mathbf{I} - \mathbf{\Phi}(\mathbf{\Phi}^\mathsf{T}\mathbf{\Phi} + \lambda\mathbf{I})^{-1}\mathbf{\Phi}^\mathsf{T} = \lambda(\mathbf{\Phi}\mathbf{\Phi}^\mathsf{T} + \lambda\mathbf{I})^{-1}\,. \tag{54}$$

**Predictions.** For the predictive equations, note that adding and subtracting $\mathbf{\Phi}\mathbf{\Phi}^\mathsf{T}\mathbf{w}_0$ to the expression for $\mathbf{w}_N$ yields:

$$
\begin{aligned}
\mathbf{w}_N &= (\mathbf{\Phi}\mathbf{\Phi}^\mathsf{T} + \lambda\mathbf{I})^{-1}(\mathbf{\Phi}\mathbf{y} + \lambda\mathbf{w}_0 + \mathbf{\Phi}\mathbf{\Phi}^\mathsf{T}\mathbf{w}_0 - \mathbf{\Phi}\mathbf{\Phi}^\mathsf{T}\mathbf{w}_0) \\
&= \mathbf{w}_0 + (\mathbf{\Phi}\mathbf{\Phi}^\mathsf{T} + \lambda\mathbf{I})^{-1}(\mathbf{\Phi}\mathbf{y} - \mathbf{\Phi}\mathbf{\Phi}^\mathsf{T}\mathbf{w}_0) \\
&= \mathbf{w}_0 + \mathbf{\Phi}(\mathbf{\Phi}^\mathsf{T}\mathbf{\Phi} + \lambda\mathbf{I})^{-1}(\mathbf{y} - \mathbf{\Phi}^\mathsf{T}\mathbf{w}_0) \,,
\end{aligned}
\tag{55}
$$

where we applied the identity $(\mathbf{I} + \mathbf{AB})^{-1}\mathbf{A} = \mathbf{A}(\mathbf{I} + \mathbf{BA})^{-1}$. Hence, letting $h_N(x) := \phi(x)^\mathsf{T}\mathbf{w}_N$, we have that:

$$
h_N(x) = h_0(x) + \phi(x)^\mathsf{T}\mathbf{\Phi}(\mathbf{\Phi}^\mathsf{T}\mathbf{\Phi} + \lambda\mathbf{I})^{-1}(\mathbf{y} - \mathbf{h}_0) \,,
\tag{56}
$$

where $\mathbf{h}_0 := \mathbf{\Phi}^\mathsf{T}\mathbf{w}_0 = [h_0(x_i)]_{i=1}^N \in \mathbb{R}^N$. In the infinite-width limit, we then have that:

$$
h_N(x) = h_0(x) + \mathbf{k}_N(x)^\mathsf{T}(\mathbf{K}_N + \lambda\mathbf{I})^{-1}(\mathbf{y} - \mathbf{h}_0) \,,
\tag{57}
$$

where we set $k := k_{\texttt{NNGP}}$ and adopt the standard GP notation for the kernel vector $\mathbf{k}_N$ and matrix $\mathbf{K}_N$.

**Underestimated variance.** Now considering $h_0 \sim \mathcal{GP}(0, k)$, we have that:

$$
\mathbb{E}[h_N(x) \mid \mathbf{y}] = \mathbf{k}_N(x)^\mathsf{T}(\mathbf{K}_N + \lambda\mathbf{I})^{-1}\mathbf{y}
\tag{58}
$$

$$
\begin{aligned}
\mathbb{V}[h_N(x) \mid \mathbf{y}] &= k(x, x) - 2\mathbf{k}_N(x)^\mathsf{T}(\mathbf{K}_N + \lambda\mathbf{I})^{-1}\mathbf{k}_N(x) \\
&\quad + \mathbf{k}_N(x)^\mathsf{T}(\mathbf{K}_N + \lambda\mathbf{I})^{-1}\mathbf{K}_N(\mathbf{K}_N + \lambda\mathbf{I})^{-1}\mathbf{k}_N(x) \\
&= k(x, x) - \mathbf{k}_N(x)^\mathsf{T}(\mathbf{K}_N + \lambda\mathbf{I})^{-1}\mathbf{k}_N(x) - \lambda\mathbf{k}_N(x)^\mathsf{T}(\mathbf{K}_N + \lambda\mathbf{I})^{-2}\mathbf{k}_N(x) \,,
\end{aligned}
\tag{59}
$$

where the last equality follows by adding and subtracting $\lambda\mathbf{I}$ from the $\mathbf{K}_N$ factor in the previous quadratic term. We can then see that the predictive variance is lower than the exact GP posterior predictive variance by a factor of $\lambda\mathbf{k}_N(x)^\mathsf{T}(\mathbf{K}_N + \lambda\mathbf{I})^{-2}\mathbf{k}_N(x)$. The two match when $\lambda \to 0$, as in Lee et al. [12]. However, for the noisy case with $\lambda > 0$, we have this mismatch, as it can also be observed in the results of Calvo-Ordoñez et al. [35]. Similarly, for the weights posterior covariance, we have that:

$$
\begin{aligned}
\widehat{\mathbf{\Sigma}}_N = \lambda^2(\mathbf{\Phi}\mathbf{\Phi}^\mathsf{T} + \lambda\mathbf{I})^{-2} &\preceq \lambda(\mathbf{\Phi}\mathbf{\Phi}^\mathsf{T} + \lambda\mathbf{I})^{-1} = \mathbf{\Sigma}_N \\
\iff \lambda(\mathbf{\Phi}\mathbf{\Phi}^\mathsf{T} + \lambda\mathbf{I})^{-2} &\preceq (\mathbf{\Phi}\mathbf{\Phi}^\mathsf{T} + \lambda\mathbf{I})^{-1} \\
\iff \lambda(\mathbf{\Phi}\mathbf{\Phi}^\mathsf{T} + \lambda\mathbf{I})^{-1} &\preceq \mathbf{I} \\
\iff (\lambda^{-1}\mathbf{\Phi}\mathbf{\Phi}^\mathsf{T} + \mathbf{I})^{-1} &\preceq \mathbf{I} \,,
\end{aligned}
\tag{60}
$$

which holds since $\mathbf{\Phi}\mathbf{\Phi}^\mathsf{T}$ is positive semidefinite and $\lambda > 0$. Hence, in the following we analyze the effect of the underestimated variance on the algorithm's regret.

**Effect on the regret bound.** We may bound the effect of the posterior variance mismatch in the regret bound of GP-TS. Let $\mathbf{\Sigma}_t = \mathbb{V}[\mathbf{w}_* | \mathbf{y}]$ represent the exact posterior covariance matrix (cf. Eq. 54) after $t \geq 1$ iterations, assuming $\lambda := \sigma_\epsilon^2$, and denote the exact and the approximate posterior, respectively, as:

$$
P_t := \mathcal{N}(\widehat{\mathbf{w}}_t, \mathbf{\Sigma}_t)
\tag{61}
$$

$$
\hat{P}_t := \mathcal{N}(\widehat{\mathbf{w}}_t, \widehat{\mathbf{\Sigma}}_t) \,.
\tag{62}
$$

Correspondingly, we set:

$$
x^* \in \operatorname*{argmax}_{x \in \mathcal{X}} f(x)
\tag{63}
$$

$$
x_t \in \operatorname*{argmax}_{x \in \mathcal{X}} h_t(x) \,,
\tag{64}
$$

assuming $f(x) = \phi(x)^\mathsf{T}\mathbf{w}_*$, for some $\mathbf{w}_* \sim \mathcal{N}(\mathbf{0}, \mathbf{I})$. The instant regret at iteration $t \geq 1$ is then:

$$\mathbb{E}[f(x^*) - f(x_t)] = \mathbb{E}[\mathbb{E}[f(x^*) - f(x_t) \mid \mathcal{D}_{t-1}]]$$

$$= \mathbb{E}\left[ \int_{\mathbb{R}^M} \int_{\mathbb{R}^M} f(x^*) - f(x_t)\,\mathrm{d}P_{t-1}(\mathbf{w}_*)\,\mathrm{d}\hat{P}_{t-1}(\mathbf{w}_t) \right]$$

$$= \mathbb{E}\left[ \int_{\mathbb{R}^M} \int_{\mathbb{R}^M} (f(x^*) - f(x_t)) \frac{\mathrm{d}\hat{P}_{t-1}}{\mathrm{d}P_{t-1}}(\mathbf{w}_t)\,\mathrm{d}P_{t-1}(\mathbf{w}_*)\,\mathrm{d}P_{t-1}(\mathbf{w}_t) \right] \quad (65)$$

$$\leq \mathbb{E}\left[ \left\| \frac{\mathrm{d}\hat{P}_{t-1}}{\mathrm{d}P_{t-1}} \right\|_\infty \int_{\mathbb{R}^M} \int_{\mathbb{R}^M} f(x^*) - f(x_t)\,\mathrm{d}P_{t-1}(\mathbf{w}_*)\,\mathrm{d}P_{t-1}(\mathbf{w}_t) \right],$$

where we applied Hölder's inequality, noting that $f(x^*) - f(x_t) \geq 0$. Therefore, if the Radon-Nikodym derivative $\frac{\mathrm{d}\hat{P}_{t-1}}{\mathrm{d}P_{t-1}}$ is uniformly bounded, the regret bound remains the same. In the finite-width case $M < \infty$, the density ratio between multivariate normal distributions with the same mean gives us:

$$\frac{\mathrm{d}\hat{P}_t}{\mathrm{d}P_t}(\mathbf{w}) = \sqrt{\frac{\det(\boldsymbol{\Sigma}_t)}{\det(\widehat{\boldsymbol{\Sigma}}_t)}} \exp\left( -\frac{1}{2}(\mathbf{w} - \widehat{\mathbf{w}}_t)^\mathsf{T}(\widehat{\boldsymbol{\Sigma}}_t^{-1} - \boldsymbol{\Sigma}_t^{-1})(\mathbf{w} - \widehat{\mathbf{w}}_t) \right), \quad \mathbf{w} \in \mathbb{R}^M. \quad (66)$$

As $\widehat{\boldsymbol{\Sigma}}_t \preceq \boldsymbol{\Sigma}_t$ (60), the difference between the inverses $\widehat{\boldsymbol{\Sigma}}_t^{-1} - \boldsymbol{\Sigma}_t^{-1}$ is positive semidefinite. The maximum is then achieved at $\mathbf{w} = \widehat{\mathbf{w}}_t$, yielding:

$$\left\| \frac{\mathrm{d}\hat{P}_t}{\mathrm{d}P_t} \right\|_\infty = \sqrt{\frac{\det(\boldsymbol{\Sigma}_t)}{\det(\widehat{\boldsymbol{\Sigma}}_t)}}$$

$$= \sqrt{\det\left( \boldsymbol{\Sigma}_t \widehat{\boldsymbol{\Sigma}}_t^{-1} \right)}$$

$$= \sqrt{\det\left( \mathbf{I} + \lambda^{-1} \boldsymbol{\Phi}\boldsymbol{\Phi}^\mathsf{T} \right)} \quad (67)$$

$$= \sqrt{\det\left( \mathbf{I} + \lambda^{-1} \boldsymbol{\Phi}^\mathsf{T}\boldsymbol{\Phi} \right)}$$

where we applied Sylvester's determinant identity to third line, and a standard determinant identity yields the last equality. In the infinite-width limit as $M \to \infty$, we have that $\boldsymbol{\Phi}^\mathsf{T}\boldsymbol{\Phi}$ converges to $\mathbf{K}_t := [k_{\text{NNGP}}(x_i, x_j)]_{i,j=1}^t$, leading us to:

$$\left\| \frac{\mathrm{d}\hat{P}_t}{\mathrm{d}P_t} \right\|_\infty = \sqrt{\det(\mathbf{I} + \lambda^{-1}\mathbf{K}_t)}. \quad (68)$$

Recall the definition of the maximum information gain [33, 59]:

$$\gamma_t := \max_{\mathcal{X}_t \subset \mathcal{X} : |\mathcal{X}_t| \leq t} \frac{1}{2} \log \det(\mathbf{I} + \lambda^{-1}\mathbf{K}_t). \quad (69)$$

If we assume that the GP information gain $\frac{1}{2}\log\det(\mathbf{I} + \lambda^{-1}\mathbf{K}_t)$ is bounded by $\gamma_t$ as above, we would then have that:

$$\left\| \frac{\mathrm{d}\hat{P}_t}{\mathrm{d}P_t} \right\|_\infty \leq \exp\gamma_t, \quad (70)$$

which is usually an unbounded term, given that $\gamma_t$ is a non-decreasing function of $t$. However, for a finite domain $|\mathcal{X}| < \infty$, we trivially have that $\gamma_t \leq \gamma_{|\mathcal{X}|}$, given that the largest finite subset $\mathcal{X}_t$ of $\mathcal{X}$ is $\mathcal{X}$ itself. Hence, in this case, the following holds:

$$\forall t \in \mathbb{N}, \quad \left\| \frac{\mathrm{d}\hat{P}_t}{\mathrm{d}P_t} \right\|_\infty \leq \exp\gamma_{|\mathcal{X}|}, \quad (71)$$

which is bounded for most practical kernels. Putting it all together, we have that:

$$\forall t \in \mathbb{N}, \quad \mathbb{E}[f(x^*) - f(x_t)] \leq r_t \exp\gamma_{|\mathcal{X}|}, \quad (72)$$

where $r_t$ represents the Bayesian regret when $x_t$ maximizes a sample from the exact GP posterior, instead of its approximation. Given that $\gamma_{|\mathcal{X}|}$ is a finite constant, the asymptotic rates for the Bayesian cumulative regret remain the same even in the presence of an underestimated predictive variance.

**Problem with $\gamma_t$ bound.** An issue with the finite bound on the Radon-Nikodym derivative above can be found when contrasting the classic definition of the maximum information gain $\gamma_t$ in the literature (69) with the actual information gain in the algorithm, i.e., the mutual information between an exact GP and the collected observations, which can be shown to be quantified by $\frac{1}{2}\log\det(\mathbf{I} + \lambda^{-1}\mathbf{K}_t)$ [59]. The issue is that, although $\gamma_t \leq \gamma_{|\mathcal{X}|}$ from the definition commonly found in the literature [31, 59], it does not necessarily follow that the actual information gain is bounded after we account for multiplicities in the eigenvalues. The algorithm is in principled allowed (and likely) to make repeated choices of the same $x_t = x_*$ for all $t$ at some point, for a fixed $x_* \in \mathcal{X}$, which may or may not be the optimizer $x^*$. In the simplest case, if all of the algorithm's choices are made at any fixed $x_* \in \mathcal{X}$, we have that:

$$
\begin{aligned}
\log\det(\mathbf{I} + \lambda^{-1}\mathbf{K}_t) &= \log\det(\mathbf{I} + \lambda^{-1}\boldsymbol{\Phi}_t^\mathsf{T}\boldsymbol{\Phi}_t) \\
&= \log\det(\mathbf{I} + \lambda^{-1}\boldsymbol{\Phi}_t\boldsymbol{\Phi}_t^\mathsf{T}) \\
&= \log\det(\mathbf{I} + t\lambda^{-1}\boldsymbol{\phi}(x^*) \otimes \boldsymbol{\phi}(x^*)) \\
&= \log\det(1 + t\lambda^{-1}\|\boldsymbol{\phi}(x^*)\|^2) \\
&\geq \log(1 + ct),
\end{aligned}
\tag{73}
$$

for some constant $c > 0$. Therefore, this lower bound diverges as $t \to \infty$, whereas $\gamma_t \leq \gamma_{|\mathcal{X}|} < \infty$ remains bounded, leading to a contradiction of the previous conclusion in Eq. 72.

**Exactly matching the posterior variance.** The exact weights posterior covariance $\boldsymbol{\Sigma}_t$ can be matched if, besides randomizing the initial weights, we randomize the observations by adding noise $\tilde{\boldsymbol{\epsilon}}_t \sim \mathcal{N}(\mathbf{0}, \lambda\mathbf{I})$ to them at training time, following a *randomize-then-optimize* approach [60]. Specifically, we minimize the perturbed loss:

$$
\tilde{\ell}_t(\mathbf{w}) := \frac{1}{2}\|\boldsymbol{\Phi}_t^\mathsf{T}\mathbf{w} - (\mathbf{y}_t + \tilde{\boldsymbol{\epsilon}}_t)\|^2 + \frac{\lambda}{2}\|\mathbf{w} - \mathbf{w}_0\|^2, \quad \tilde{\boldsymbol{\epsilon}}_t \sim \mathcal{N}(\mathbf{0}, \lambda\mathbf{I}), \quad \mathbf{w}_0 \sim \mathcal{N}(\mathbf{0}, \mathbf{I}). \tag{74}
$$

The corresponding minimizer is given by:

$$
\tilde{\mathbf{w}}_t = (\boldsymbol{\Phi}_t\boldsymbol{\Phi}_t^\mathsf{T} + \lambda\mathbf{I})^{-1}(\boldsymbol{\Phi}_t(\mathbf{y}_t + \tilde{\boldsymbol{\epsilon}}_t) + \lambda\mathbf{w}_0), \tag{75}
$$

whose conditional mean still matches the exact weights posterior mean:

$$
\mathbb{E}[\tilde{\mathbf{w}}_t \mid \mathbf{y}_t] = (\boldsymbol{\Phi}_t\boldsymbol{\Phi}_t^\mathsf{T} + \lambda\mathbf{I})^{-1}\boldsymbol{\Phi}_t\mathbf{y}_t \tag{76}
$$

and whose conditional covariance now satisfies:

$$
\begin{aligned}
\mathbb{V}[\tilde{\mathbf{w}}_t \mid \mathbf{y}_t] &= \mathbb{V}[(\boldsymbol{\Phi}_t\boldsymbol{\Phi}_t^\mathsf{T} + \lambda\mathbf{I})^{-1}(\boldsymbol{\Phi}_t(\mathbf{y}_t + \tilde{\boldsymbol{\epsilon}}_t) + \lambda\mathbf{w}_0) \mid \mathbf{y}_t] \\
&= (\boldsymbol{\Phi}_t\boldsymbol{\Phi}_t^\mathsf{T} + \lambda\mathbf{I})^{-1}\mathbb{V}[\boldsymbol{\Phi}_t\tilde{\boldsymbol{\epsilon}}_t + \lambda\mathbf{w}_0](\boldsymbol{\Phi}_t\boldsymbol{\Phi}_t^\mathsf{T} + \lambda\mathbf{I})^{-1} \\
&= (\boldsymbol{\Phi}_t\boldsymbol{\Phi}_t^\mathsf{T} + \lambda\mathbf{I})^{-1}(\lambda\boldsymbol{\Phi}_t\boldsymbol{\Phi}_t^\mathsf{T} + \lambda^2\mathbf{I})(\boldsymbol{\Phi}_t\boldsymbol{\Phi}_t^\mathsf{T} + \lambda\mathbf{I})^{-1} \\
&= \lambda(\boldsymbol{\Phi}_t\boldsymbol{\Phi}_t^\mathsf{T} + \lambda\mathbf{I})^{-1} \\
&= \boldsymbol{\Sigma}_t,
\end{aligned}
\tag{77}
$$

thereby recovering the exact posterior covariance. Nevertheless, note that the original difference in posterior predictive variance according to Eq. 59 is $\lambda\mathbf{k}_t(x)^\mathsf{T}(\mathbf{K}_t + \lambda\mathbf{I})^{-2}\mathbf{k}_t(x)$, which is typically negligible for small values of noise variance $\sigma_\epsilon^2 = \lambda$. As a consequence, our cumulative regret bound remains approximately valid for NOTS, despite the slight underestimation of the posterior variance.

## D  Experiment details

### D.1  Darcy flow

Darcy flow describes the flow of a fluid through a porous medium with the following PDE form

$$
\begin{aligned}
-\nabla \cdot (a(x)\nabla u(x)) &= g(x), \quad x \in \Omega = (0,1)^2 \\
u(x) &= 0, \quad x \in \partial\Omega,
\end{aligned}
$$

where $u(x)$ is the flow pressure, $a(x)$ is the permeability coefficient and $g(x)$ is the forcing function. We fix $g(x) = 1$ and generate different solutions at random with zero Neumann boundary conditions

on the Laplacian, following the setting in Li et al. [28], as implemented by the neural operator package [47]. In particular, for this problem, we generate a search space $\mathcal{S}$ with $|\mathcal{S}| = 1000$ data points. The divergence of $f$ is $\nabla \cdot f = \frac{\partial f_x}{\partial x} + \frac{\partial f_y}{\partial y}$ where $f : \Omega \to \mathbb{R}^2$ is a vector field $f = (f_x, f_y)$. The gradient $\nabla u = (\frac{\partial u(x,y)}{\partial x}, \frac{\partial u(x,y)}{\partial y})$ where $u(x,y) : \Omega \to \mathbb{R}$ is a scalar field. Inspired by previous works [5, 50, 61], we chose the following objective functions to evaluate the functions assuming that we aim to maximize the objective function $f(\cdot)$:

1. Negative total flow rates [50]

$$f(u, a) = \int_{\partial \Omega} a(x)(\nabla u(x) \cdot n)ds$$

where $s = \partial \Omega$ is the boundary of the domain and $n$ is the outward pointing unit normal vector of the boundary. $q(x) = -a(x)\nabla u(x)$ is the volumetric flux which describes the rate of volume flow across a unit area. Therefore, the objective function measures the boundary outflux. Since the boundary is defined on a grid, $n \in \{[-1, 0], [1, 0], [0, 1], [0, -1]\}$ for the left, right, top and bottom boundaries. The boundary integral can be simplified as

$$\int_0^1 [-a(0, y)u_x(0, y) + a(1, y)u_x(1, y)]dy + \int_0^1 [-a(x, 0)u_y(x, 0) + a(x, 1)u_y(x, 1)]dx$$

where $u_x(x, y) = \frac{\partial u}{\partial x}, u_y(x, y) = \frac{\partial u}{\partial y}$

2. Negative total pressure (Eq 2.1 in [51])

$$f(u, g) = -\frac{1}{2} \int_\Omega (\|u(x)\|_2 + \beta \|g(x)\|_2)dx$$

with $\beta > 0$ is a coefficient for the forcing term $g(x)$. With a constant $g(x)$, the objective is simplified as $-\frac{1}{2} \int_\Omega \|u(x)\|_2 dx$.

3. Negative total potential energy [5]

$$f(u, a) = -\int_{\mathcal{X}} a(x)\|\nabla u(x)\|^2 \, dx + \int_{\mathcal{X}} s(x)u(x) \, dx$$

This functional corresponds to the system's total potential energy. It balances the energy dissipated by fluid friction (the first term) against the potential energy supplied by the uniform fluid source (the second term, where $s = 1$ is assumed). In our design optimization context, where the underlying physical state $u$ is already a stable solution to the Darcy PDE, minimization of this functional over the set of permeability fields $a \in \mathcal{S}$ determines the permeability field $a^*$ that requires the minimum total energy to sustain the required fluid injection (source $s = 1$) while maintaining zero pressure at the boundary ($u = 0$). This effectively identifies the most hydrodynamically efficient design for the given flow constraints. This functional is related to the potential power functional in Wiker et al. [5] with the difference that the latter requires estimates of the velocity field, while the simplified energy calculation above only uses the pressure field $u$.

## D.2 Shallow Water

The shallow water equation on the rotating sphere is often used to model ocean waters over the surface of the globe. This problem can be described by the following PDE [46]:

$$\frac{\partial \varphi}{\partial t} + \nabla \cdot (\varphi v) = 0 \quad \text{in} \quad \mathbb{S}^2 \times \{0, +\infty\}$$

$$\frac{\partial (\varphi v)}{\partial t} + \nabla \cdot (\varphi v \otimes v) = g \quad \text{in} \quad \mathbb{S}^2 \times \{0, +\infty\}$$

$$\varphi = \varphi_0, \quad v = v_0 \quad \text{on} \quad \mathbb{S}^2 \times \{0\}$$

where the input function is defined as the initial condition of the state $a = (\varphi_0, \varphi_0 v_0)$ with the geopotential layer depth $\varphi$ and the discharge ($v$ is the velocity field), $g$ is the Coriolis force term, and

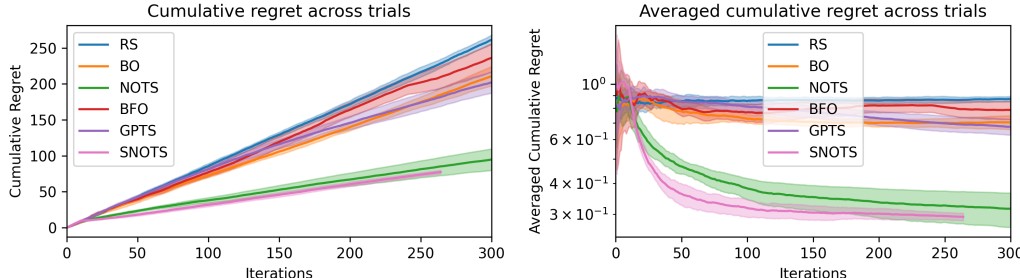

Figure 4: Cumulative regret across trials for the Darcy *flow rate* optimization problem with only the last linear layer of a single-hidden-layer FNO trained via full-batch gradient descent for NOTS (labeled as SNOTS). All our results were averaged over 10 independent trials, and shaded areas represent $\pm 1$ standard deviation.

$\mathbb{S}^2$ denotes the surface of the 2-sphere in $\mathbb{R}^3$. The output function $u$ predicts the state function at time $t$: $(\varphi_t, \varphi_t v_t)$. For this problem, we use a search space $\mathcal{S}$ with $|\mathcal{S}| = 200$ data points.

As the shallow water equation is usually chosen as a simulator of global atmospheric variables, we adopt the most common data assimilation objective [52, 53] in the weather forecast literature defined as:

$$f(u, a) = \frac{1}{2}\langle a - a_p, B^{-1}(a - a_p)\rangle + \frac{1}{2}\langle u - u_t, R^{-1}(u - u_t)\rangle,$$

where $a_p$ describes the prior estimate of the initial condition, $u_t$ represents the ground truth function, the background kernel $B$ and error kernel $R$ can be computed with historical data. The objective can be defined as an inverse problem which corresponds to finding the initial condition $a$ that generates the ground truth solution function $u_t$. Here we simplify the objective by not penalizing the initial condition (dropping the prior term) and assuming independence and unit variance on the solution functions using an identity kernel $R$), the simplified objective function $f(u) = \frac{1}{2}\langle u - u_t, u - u_t\rangle$ can be used to measure different initial conditions.

### D.3 Noise

To simulate real-world settings, noise was added to the observations by computing the empirical covariance matrix of the outputs $y$ in the dataset for the corresponding PDE and then adding Gaussian noise with variance set to 1% of the coordinate-wise output variance.

### D.4 Algorithm settings

NOTS was implemented using the *Neural Operator* library [47] and run on NVIDIA H100 GPUs on CSIRO's high-performance computing cluster. For each dataset, we selected the recommended settings for FNO models according to examples in the library. Parameters were randomly initialized using Kaiming (or He) initialization [36] for the network weights, sampling from a normal distribution with variance inversely proportional to the input dimensionality of each layer, while biases were initialized to zero. For all experiments, we trained the model for 10 epochs of mini-batch stochastic gradient descent with an initial learning rate of $10^{-3}$ and a cosine annealing scheduler. The regularization factor for the L2 penalty was set as $\lambda := 10^{-4}$. This same setting for the regularization factor was also applied to our implementation of STO-NTS.

## E  Additional results with single-hidden-layer model

More closely to the setting in our theoretical results, we tested a single-hidden-layer FNO on the Darcy flow PDE. Only the last hidden layer of the model was trained via full-batch gradient descent. The FNO was configured without any lifting layer, having only a single Fourier kernel convolution and a residual connection, as in the original formulation. The number of hidden channels was set to 2048 to approximate the infinite-width limit.

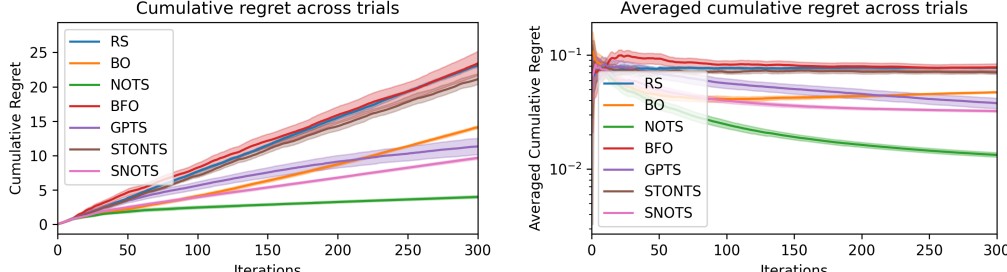

Figure 5: Cumulative regret across trials for the Darcy flow *total pressure* optimization problem with only the last linear layer of a single-hidden-layer FNO trained via full-batch gradient descent for NOTS (labeled as SNOTS).

The results in Figure 4 show that the algorithm with the simpler model (SNOTS) can perform well in this setting, even surpassing the performance of the original NOTS. However, in the more challenging scenario imposed by the potential power problem [adapted from 5], we note that SNOTS struggles, only achieving mid-range performance when compared to other baselines, as shown in Figure 5. This performance drop suggests that the complexity of the pressure optimization problem may require more accurate predictions to capture details in the output functions that might heavily influence the potential power. In general, a quadratic objective will be more sensitive to small disturbances than a linear functional, hence requiring a more elaborate model.

## F    Limitations and extensions

**Noise.**    We note that, although our result in Proposition 2 assumes a well specified noise model, it should be possible to show that the same holds for noise which is sub-Gaussian with respect to the regularization factor. The latter would allow for configuring the algorithm with any regularization factor which is at least as large as the assumed noise sub-Gaussian parameter (i.e., its variance if Gaussian distributed). However, this analysis can be quite involved and out of the immediate scope of this paper. Therefore, we leave such investigation for further research.

**Nonlinear functionals.**    We assumed a bounded linear functional in Proposition 2, which should cover a variety of objectives involving integrals and derivatives of the operator's output. However, this assumption may not hold for more interesting functionals, such as some objectives considered in our experiments. Similar to the case with noise, any Lipschitz continuous functional of the neural operator's output should follow a sub-Gaussian distribution [62]. Hence, the Gaussian approximation remains reasonable, though a more in-depth analysis would be needed to derive the exact rate of growth for the cumulative regret in these settings.

**Mult-layer models.**    For the theoretical analysis, we assumed a single hidden layer neural network as the basis of our Thompson sampling algorithm. While this choice provides a simple and computationally efficient framework, it may not be optimal for all applications or datasets. For instance, in some cases, a deeper neural network with more layers might provide better performance due to increased capacity to capture complex patterns in the data. Extending our analysis to this setting involves extending the inductive proofs for the multi-layer NNGP [38, 54] to the case of neural operators. Such extension, however, may require transforming the operator layer's output back into a function in an infinite-dimensional space, which may lead to a bottleneck effect affecting the possibility of a kernel limit [55]. In the single-hidden-layer case, such effect is avoided by operating directly with the finite-dimensional input function embedding $A_{\mathbf{R}}(a)(z) \in \mathbb{R}^{d_{\mathbf{R}}}$. Recently, concurrent work has explored the infinite-width limit for multi-layer neural operators [63], but their applicability to NOTS is left as subject of future work.

**Prior misspecification.**    We assumed that the true operator $G_*$ follows the same prior as our model, which was also considered to be infinitely wide. While this assumption greatly simplifies our analysis, more practical results may be derived by considering finite-width neural operators and a true operator

which might not exactly correspond to a realization of the chosen class of neural operator models. For the case of finite widths, one simple way to obtain a similar regret bound is to let the width of the network grow at each Thompson sampling iteration. The approximation error between the GP model and the finite width neural operator can potentially be bounded as $\mathcal{O}(M^{-1/2})$ [55]. Hence if the sequence of network widths $\{M_t\}_{t=1}^{\infty}$ is such that $\sum_{t=1}^{\infty} \frac{1}{\sqrt{M_t}} < \infty$, a similar regret bound to the one in Proposition 2 should be possible. Furthermore, if other forms of prior misspecification need to be considered, analyzing the Bayesian cumulative regret (instead of the more usual frequentist regret), as we did, allows one to bound the resulting cumulative regret of the misspecified algorithm via the Radon-Nikodym derivative $\frac{\mathrm{d}P}{\mathrm{d}\hat{P}}$ of the true prior $P$ with respect to the algorithm's prior probability measure $\hat{P}$. If its essential supremum $\left\| \frac{\mathrm{d}P}{\mathrm{d}\hat{P}} \right\|_{\infty}$ is bounded, then the resulting cumulative regret remains proportional to the same bound derived as if the algorithm's prior was the correct one [8].

# G   Broader impact

This work primarily focuses on the theoretical exploration of extending Thompson sampling to function spaces via neural operators. As such, it does not directly engage with real-world applications or present immediate societal implications. However, the potential impact of this research lies in its application. By advancing methods for function-space optimization, this work may indirectly contribute to various fields that utilize complex simulations and models, such as climate science, engineering, and physics. Improvements in computational efficiency and predictive power in these fields could lead to positive societal outcomes, such as better climate modeling or engineering solutions. Nevertheless, any algorithm with powerful optimization capabilities carries ethical considerations. Its deployment in domains with safety-critical implications must be approached with care to avoid misuse or unintended consequences. Researchers and practitioners should ensure transparency, fairness, and accountability in applications potentially affecting society.

