# OpenReview forum: "Thompson Sampling in Function Spaces via Neural Operators"
_NeurIPS.cc/2025/Conference — NeurIPS 2025 poster_

### Official Review · Reviewer_aMDd · 2025-06-29

**Clarity:** 3
**Significance:** 2
**Originality:** 3
**Rating:** 4
**Confidence:** 2

**Summary:**

The paper studies the problem on black-box optimization of a function composed of a known functional with a black-box unknown operator. It proposes a method based on neural operator and Thompson sampling.

**Questions:**

1. It is assumed that the search space is compact. Does it mean that essentially the searched function is parameterized.
2. In algorithm 1, I feel the notation $\arg\min$ is a bit unsuitable. Indeed, you can usually not gurantee training it to global optimal with neural network due to non-convex optimization.

**Ethical Concerns:**

["NO or VERY MINOR ethics concerns only"]

**Limitations:**

It is not so clear to me that why neural network should be the solution for this problem, since it can be computationally expensive and requires lots of data.

**Paper Formatting Concerns:**

None.

**Quality:**

3

**Strengths And Weaknesses:**

Strengths: The paper is overall well-written and clear. The studied problem is important and interesting. The proposed solution has theoretical guarantee. Experimental results show that it outperforms other baseline methods.
Weaknesses: From my understanding, training a neural network usually requires lots of data and computation resource. But black-box optimization typically has limited data. Then what is the advantage of using NN instead of, e.g., Gaussian process based Thompson sampling.

---

> ### Author Rebuttal · Authors · 2025-07-31
>
> We'd like to thank the reviewer for their positive feedback and insightful comments.  We address the reviewer's questions below.
>
> ### Weaknesses / Limitations
> **NN vs. Gaussian Process.**
> "what is the advantage of using NN instead of, e.g., Gaussian process based Thompson sampling." And states: "It is not so clear to me that why neural network should be the solution for this problem, since it can be computationally expensive and requires lots of data."
>
> **Our Response:** Although GPs typically perform well for Bayesian modeling tasks with low volumes of data, we have high-dimensional data in both inputs and outputs of the model, which render the application of traditional multi-output GP models challenging.  For the shallow water PDE, for example, both inputs and outputs lie in 6144-dimensional space.  Hence, without specialized kernels to capture spatial data on both inputs and outputs, conventional models would struggle due to both the challenges of modeling high-dimensional input data and the elevated computational cost of vector-valued GP inference with very large number of outputs.  In contrast, neural operators are specially designed to deal with function-valued input and output data, typically over spatial domains.  In addition, their training cost only grows linearly with the number of data points, instead of cubically (as in GPs).  Hence, NOTS is more scalable to accommodate longer runs or extensions with batched evaluations than traditional GP solutions, though we considered only up to 300 iterations in our experiments to allow for comparisons against GP baselines.
>
> ### Questions
>
> **Question 1: Compact Search Space.**
> "It is assumed that the search space is compact.  Does it mean that essentially the searched function is parameterized?"
>
> **Our Response:** Some sort of parameterization is required to allow a computational method to search through the space.  In our case, we simply assumed the input functions and the observations are discretized on a grid.  Compactness then simply means that the values the functions can take on the grid cells are bounded.  Note, however, that this is always the case when the search space is finite.
>
> **Question 2: Notation in Algorithm 1.**
> "In algorithm 1, I feel the notation is a bit unsuitable.  Indeed, you can usually not guarantee training it to global optimal with neural network due to non-convex optimization.  arg min"
>
> **Our Response:** In Algorithm 1, we use the $\arg\min$ notation to denote the minimizer obtained by gradient descent;  we will add a remark on this relaxed use of the notation.  Nonetheless, although optimization landscapes for neural networks are usually non-convex, plenty of recent works have shown that, in the overparameterized regime, (stochastic) gradient descent converges to global minima (Allen-Zhu et al., 2019).  In addition, in the infinite-width limit, the loss function is a convex quadratic objective with respect to the network parameters, leading to a unique global minimum.
>
> ### References
> * Allen-Zhu, Z., Li, Y., & Song, Z. (2019).  A convergence theory for deep learning via over-parameterization. Proceedings of the 36th International Conference on Machine Learning (ICML 2019).
> * Jeong, S., & Lee, S. (2025). Optimal control for Darcy's equation in a heterogeneous porous media.  Applied Numerical Mathematics, 207, 303-322.
> * Rabier, F., Thépaut, J. N., & Courtier, P. (1998).  Extended assimilation and forecast experiments with a four‐dimensional variational assimilation system. Quarterly Journal of the Royal Meteorological Society, 124(550), 1861-1887.

---

> > ### Author Response · Authors · 2025-08-09
> > **Follow up**
> >
> > Dear Reviewer aMDd,
> >
> > Thanks again for your feedback. Given the approaching author-reviewer discussion deadline, we are wondering if our response has fully addressed the Reviewer's concerns or if there are any further points to clarify.

---

### Official Review · Reviewer_9PFn · 2025-07-01

**Clarity:** 2
**Significance:** 3
**Originality:** 3
**Rating:** 4
**Confidence:** 2

**Summary:**

The paper presents an extension of Thompson sampling for use in neural operators. The proposed method avoids the computation of uncertainty. It is connected to active learning via sampling more effective data points which are used to train neural surrogates to solve PDEs. Experiments are performed on Darcy flow and shallow water modelling problems, via their respective PDEs. It is claimed that the proposed method (NOTS) helps to encode the compositional structure of problems and improves performance.

**Questions:**

1) Can authors clarify what is the main goal/ motivation behind this work?

2) Where exactly is Thompson sampling coming into play? It feels like using a term without any necessity.

3) Since, the goal of the paper is not clear; hence the evaluation metric presented is also not justified. The only metric of comparison is cumulative regret. Based on equation (16) the computation of such a quantity requires the use of a*. What is used to compute this metric?

4) What is the set (S) over which the optimization is performed to choose a_t? How is this chosen?

5) What is the rationale behind the choice of optimization functionals?

6) Please explain and discuss the results in the paper more thoroughly, it will help appreciate the method. Can you introduce Thompson sampling in Section 2?

**Ethical Concerns:**

["NO or VERY MINOR ethics concerns only"]

**Final Justification:**

Based on the authors rebuttal which has satisfied my concerns, I have updated the score

**Limitations:**

Limitations have been discussed.

**Paper Formatting Concerns:**

No major formatting issue.

**Quality:**

2

**Strengths And Weaknesses:**

Strength:

1. The relevant literature has been discussed thoroughly.
2. The problem formulation including neural PDE solvers and Gaussian process are introduced sufficiently.
3. Theoretical results claimed in the paper are reasonable.

Weakness:

1. The point of the paper is absolutely unclear. The problem and the proposed solution have not been motivated appropriately.
2. The evaluation of the method is not convincing.
3. One of the main components of the proposed method are the optimization functional, the have been used in an ad-hoc manner. There is no justification and they vary from one problem to another, without any context.

Please see questions for more details.

---

> ### Author Rebuttal · Authors · 2025-07-31
>
> We would like to thank the reviewer for the constructive feedback and helpful criticism of our paper.  We address the main concerns in our response below.
>
> ### Weaknesses
>
> **Motivation & Problem Formulation.**
> "The point of the paper is absolutely unclear.  The problem and the proposed solution have not been motivated appropriately."
>
> **Evaluation.**
> "The evaluation of the method is not convincing."
>
> **Optimization Functionals.**
> "One of the main components of the proposed method are the optimization functional, the have been used in an ad-hoc manner.  There is no justification and they vary from one problem to another, without any context."
>
> ### Questions
>
> **Question 1: Main Goal/Motivation.**
> "Can authors clarify what is the main goal/ motivation behind this work?"
>
> **Our Response:**
> One of our main motivations was to propose a method to optimize functionals involving simulation outputs.  With the rise of neural operators, we now have access to powerful data-driven emulators that can accurately mimic PDE solvers.  Thompson sampling then becomes an appealing approach to include such ML models within a Bayesian optimization framework, given the results in NTK theory and related areas studying overparameterized neural networks.  In the standard neural operator setting, we can only predict the output solution given the input, but we may have no idea about which input would achieve the best objective of interest, and this question can be answered by Thompson sampling in the Bayesian optimization framework.  For instance, in our Darcy flow example in section 6.2, a good neural operator can precisely predict the output flow given the input permeability function, but we still do not know which permeability function results in the minimum sinkage (our objective), and our framework indicates the best possible input function (optimal permeability function) to achieve this objective (minimum sinkage).  Lastly, we note that, in general, the objective functional will be problem dependent, and we assume that the most expensive step is the forward simulation which the neural operator is learning to emulate, not the evaluation of the objective functional on the simulated/emulated output.
>
> **Question 2: Role of Thompson Sampling.**
> "Where exactly is Thompson sampling coming into play?  It feels like using a term without any necessity."
>
> **Our Response:**
> We show that training a neural operator from randomly initialized weights is approximately equivalent to Thompson sampling, which is a well known algorithm in the Bayesian optimization (BO) and multi-armed bandits literature, allowing us to derive theoretical performance guarantees for NOTS.  The start of our section 3.2 is reference [25], and therein the introduction gives a description of Thompson sampling that is relevant to our context.  We will include a similar introduction in our supplementary material.  A more classical introduction to Thompson sampling can be found in the works of Russo and Van Roy [42].
>
> Classic Thompson sampling, as other algorithms for BO, can be seen as trying to estimate the (global) optimum $x^* \in \arg\max_{x\in\mathcal{X}} f(x)$ of a (black-box) function $f:\mathcal{X}\to\mathbb{R}$ within a limited budget of (noisy) evaluations.  In the simple case of Bayesian linear regression surrogates $\hat{f}_t(x) = \mathbf{w}_t ^\top \phi(x)$, $\mathbf{w} _t \in \mathbb{R}^M$, for a scalar-valued objective $f: \mathcal{X} \to \mathbb{R}$, we assume $f(x) = \mathbf{w} ^\top \phi(x)$, for some $\mathbf{w} \sim \mathcal{N}(\mathbf{0}, \mathbf{I})$, Thompson sampling proceeds as follows, for $t \in \{1, \dots, T\}$:
> 1.  Sample $\mathbf{w}_t \sim N( \mathbf{\hat{w}}, \Sigma )$, where $\mathbf{\hat{w}} := \mathbb{E}[\mathbf{w} | D ]$, $\Sigma := \mathbb{V}[\mathbf{w} | D]$;
> 2.  Optimize $x_t \in \arg\max_{x \in \mathcal{X}} \mathbf{w}_t^\top \boldsymbol{\phi}(x)$;
> 3.  Observe $y_t = f(x_t) + \epsilon_t$;
> 4.  Update the observations dataset $D \leftarrow D \cup \lbrace x_t, y_t \rbrace$;
> 5.  Increment $t \leftarrow t + 1$, and repeat while $t \leq T$;
>
> where $\mathbf{\hat{w}}$ and $\Sigma$ represent the posterior mean vector and covariance matrix, respectively, and $D$ denotes the set of observations.  The main difference with respect to NOTS and neural Thompson sampling (NTS) algorithms is that step 1 is replaced with optimization of a regularized least-squares loss (Eq. 9 for NOTS) via gradient descent, which yields an approximate posterior sample.  In STO-NTS, the feature map $\phi: \mathcal{X} \to \mathbb{R}^M$ above also has a natural correspondence with the output of the last hidden layer of a randomly initialized neural network, which has only the readout (last) layer trained.
>
> **Question 3: Justification of Cumulative Regret.**
> "Since, the goal of the paper is not clear;  hence the evaluation metric presented is also not justified. The only metric of comparison is cumulative regret.  Based on equation (16) the computation of such a quantity requires the use of a\*.  What is used to compute this metric?"
>
> **Our Response:**
> Regret is the standard choice of metric in the Thompson sampling, bandits and Bayesian optimization literature to analyze the theoretical performance of an optimization algorithm.  It is usually not possible to directly measure regret in practical applications, as we might not have access to the true optimum of the objective, as the reviewer has correctly pointed out.  However, for the purposes of this paper, and similar to other BO/bandits papers, our examples with synthetic data allow us to directly measure regret for performance comparisons, since we have access to the true noise-free functional and can evaluate it over the entire search space. In the revised version of the paper, we will clarify the role of regret and include additional non-regret plots (e.g., the maximum value observed up to each iteration), which might be more meaningful for readers less familiar with the bandits/BO literature.
>
> **Question 4: Optimization set $\mathcal{S}$.**
> "What is the set $\mathcal{S}$ over which the optimization is performed to choose a\_t?  How is this chosen?"
>
> **Our Response:**
> The set $\mathcal{S}$ represents the search space which the algorithm has access to.  It is a subset of the domain $\mathcal{A}$ of the true (unknown) operator $G*$ , domain which is shared with the neural operators $G^*$.  For instance, in the Darcy flow experiments, the domain $\mathcal{A}$ is the set of all functions mapping the 2D unit box $[0,1]^2$ to the non-negative real line $\mathbb{R}_+$, whereas the search space $\mathcal{S}$ is a random finite subset of the domain, generated by sampling binary values over 2D grids within the unit square.
>
> **Question 5: Rationale for Optimization Functionals.**
> "What is the rationale behind the choice of optimization functionals?"
>
> **Our Response:**
> Our framework is agnostic to the choice of objective functional, which could even be, e.g., application-specific black-box code run on top of a simulator output, as long as these functionals can be (cheaply) evaluated on the output of the true operator (i.e., the simulator or real process we are trying to emulate) and on the output of the neural operator.  Given the specific modeling goal of a PDE, the objective can vary.  For example, for the Darcy flow problem, one of our objectives is to find an optimal topological design of the input function which minimizes the total sinkage (Jeong, S. et al., 2025).  We therefore adopted the objective in the literature to measure the total pressure.  Contrastingly, for the shallow water equation problem, we wonder how to trace back the initial condition of a specific output function, e.g., a hurricane, as this PDE represents a weather forecast problem (Rabier, F. et al, 1998), and we therefore formulated an inverse problem as the objective.
>
> **Question 6: Deeper Explanation of Results.**
> "Please explain and discuss the results in the paper more thoroughly, it will help appreciate the method.  Can you introduce Thompson sampling in Section 2?"
>
> **Our Response:**
> With the additional page allowed for the camera-ready version of the paper, we plan to add a brief overview of Thompson sampling in the background section (Sec. 2) and a more in-depth discussion of the experimental results.  In particular, we will extend the discussion on the results of Fig. 2 and 3, highlighting that GP-TS and BO with the log-expected improvement acquisition function, both equipped with the infinite-width BNN kernel, present improved performance in comparison with the other baselines, confirming the observations of Li et al.  [40] that this choice of GP covariance function yields competitive results in high-dimensional settings.

---

> > ### Author Response · Authors · 2025-08-01
> > **Missing references**
> >
> > References Jeong et al. (2015) and Rabier et al. (1998) can be found in the rebuttal to Reviewer aMDd.

---

> > ### Comment · Reviewer_9PFn · 2025-08-04
> >
> > Dear authors,
> >
> > Thanks for the response. It has clarified the purpose of the paper. I will strongly suggest that the example mentioned in the Rebuttal or something similar be highlighted in the paper, the reason being although the paper deals with neural operators, the end of goal of the proposed method is somewhat different, this should be very clear to any reader. For other question, things are much better now, please incorporate your answers in the final version of the paper. I will upgrade my recommendation accordingly

---

### Official Review · Reviewer_28oU · 2025-07-01

**Clarity:** 4
**Significance:** 3
**Originality:** 3
**Rating:** 4
**Confidence:** 2

**Summary:**

The paper proposes Neural operator Thompson sampling, a novel method for applications which require optimization wrt an unknown functional operator which may be too expensive to evaluate using simulation. To do so, a neural operator is trained from a number of samples. The resulting operator is treated as the correct posterior consistent with the input data and is sampled to generate new input functions. The method uses the fact that for wide neural operators these samples behave like samples from Gaussian processes.

**Questions:**

Having to retrain seems like a major limitations. How do other methods compare here?

**Ethical Concerns:**

["NO or VERY MINOR ethics concerns only"]

**Final Justification:**

The authors were able to answer my questions and concerns. However as I have to admit that I am not particularly knowledgable about this domein, I have decided to lower my  confidence.

**Limitations:**

Yes

**Quality:**

3

**Strengths And Weaknesses:**

**Strengths**

- introduces a novel method for Thompson sampling using Neural Operators
- this method avoids explicit uncertainty quantification, by random reinitialization and training which acts like sampling from the posterior
- theoretical guarantees provide convergence and regret bounds in the infinite width limit
- good evaluation with baselines, although I am unfamiliar with this subdomain to assess whether these are the right baselines

**Weaknesses**

- theoretical guarantees are limited to discretized input domains
- retraining the operator at each step from scratch is expensive
- limited evaluation on toy examples

---

> ### Author Rebuttal · Authors · 2025-07-31
>
> We'd like to thank the reviewer for their valuable feedback. Below we address the issues raised.
>
> ### Weaknesses
>
> **Theoretical Guarantees.**
> "theoretical guarantees are limited to discretized input domains"
>
> **Our Response:**
> Besides simplifying some aspects of the derivation of regret bounds, we attached our theoretical analysis for Proposition 2 to the setting where function-valued inputs are evaluated over discretized domains as that is usually the case with neural operator models, especially FNO.  One cannot implement truly infinite-dimensional input/output data on a computer, leading to a need for discretization or parameterization via a finite number of parameters, so that the observation data are finite dimensional.  We, highlight, however, that the result in Proposition 1 (GP limit) still holds for infinite-dimensional inputs and outputs.
>
> **Computational Cost.**
> "retraining the operator at each step from scratch is expensive"
>
> **Our Response:**
> Answered below.
>
> **Evaluation Scope.**
> "limited evaluation on toy examples"
>
> **Our Response:**
> The scope of our experiments is in line with similar accepted works in the machine learning literature [16,25,29].  Future investigations into data from other literature, such as in [15], are possible.
>
> ### Questions
> **Question 1: Comparison of Retraining Cost.**
> "Having to retrain seems like a major limitations.  How do other methods compare here?"
>
> **Our Response:** The retraining of the neural network (NN) model from scratch is a common implicit requirement in other NN-based Thompson sampling baselines, such as STO-NTS and the original neural Thompson sampling.  This is to ensure we have a new posterior sample at every iteration, which is part of the algorithmic design for Thompson sampling.  If training only the readout layer of the neural network, as in our theoretical analysis, one could, however, keep the other layers fixed at an initial random initialization kept throughout iterations, similar to other random feature models, such as sparse spectrum Gaussian processes;  training thus requires just regularized linear regression. In this case, one can perform incremental rank-1 updates to the model weights probability distribution (Gijsberts & Metta, 2003) without having to retrain the whole model from scratch.  However, keeping the untrained layers fixed, also limits the range of the posterior samples, reducing the algorithm's exploration, leading to worse performance, especially in practical finite-width settings where the hidden layers might not be wide enough to get the model closer to GP behavior.
>
> ### References
> * Gijsberts, A., & Metta, G. (2013).  Real-time model learning using Incremental Sparse Spectrum Gaussian Process Regression. Neural Networks, 41, 59–69.  \url{[https://doi.org/10.1016/j.neunet.2012.08.011](https://doi.org/10.1016/j.neunet.2012.08.011)}

---

> ### Comment · Reviewer_28oU · 2025-08-05
>
> I thank the authors for their rebuttal. While I have to admit that I am not particularly familiar with this domain, I feel that an example which goes beyond the toy examples would greatly benefit this paper. For this reason I choose to retain my score but I do recommend the AC to disregard my review in case of doubt given my low familiarity with the domain.

---

> > ### Author Response · Authors · 2025-08-09
> >
> > We would like to thank Reviewer 28oU again for their valuable feedback. We are working on additional examples to include in the paper's revision, and we will make sure to incorporate the remaining feedback on the discussion of theoretical results and computational costs.

---

### Official Review · Reviewer_Gc9e · 2025-07-02

**Clarity:** 2
**Significance:** 2
**Originality:** 2
**Rating:** 4
**Confidence:** 4

**Summary:**

The paper proposed the Neural Operator Thompson Sampling (NOTS) Algorithm to solve the optimization problem in function space. The specific optimization objective is the output of a known functional of an unknown operator. The main tool in use is the neural network Gaussian process (or conjugate kernel). More specifically, the paper uses the famous result that neural networks with a single hidden layer with infinite width acts like a Gaussian process. The authors provide the regret bound adapted from the existing result on Gaussian process Thompson sampling under the assumption of linear functional and a single-hidden-layer neural network. Besides, experimental results on 2D Darcy flow and a shallow water model are presented to show the advantage of the proposed NOTS algorithm.

**Questions:**

1. The good flow potential figure also has open porus in the upper right and lower right corner. Can the authors provide some explanations to it?

2. Could you rephrase line 195 to line 196. It is hard to understand.

**Ethical Concerns:**

["NO or VERY MINOR ethics concerns only"]

**Final Justification:**

The authors have addressed my concerns during the discussion. I recommend the paper to be accepted tentatively. I am unsure about the originality of the paper since the techniques in use are not that novel.

**Limitations:**

Yes

**Paper Formatting Concerns:**

Line 230: typo. deadline -> baseline

Line 194: the authors use the wording 'as previously discussed' without specifying the exact location where the result was given before. This makes the paper hard to follow.

**Quality:**

3

**Strengths And Weaknesses:**

Strength:

The author utilize neural operator to extend the theory to more practical problem with high dimension or infinite dimension. The structure of the paper is good, and the literature review is thorough. The list of benchmark algorithm is relatively complete.

Weakness:
Some parts of the proof and discussion lack clarity:
   1. Proposition C.1 and Proposition 1 are not consistent.
   2. It is unclear what neural network structure is used in the simulation of NOTS algorithm in the main paper, as the appendix specify the simulation is run under the single hidden layer NN.
   3. The simulation is not quite reproducible as some basic simulation setup is unclear. Also, in some of the figures, the simulation has fewer iterations, which I think deserve an explanation.
   4. In line 288, it says that 'STO-NTS thrives', but in Figure 2, STO-NTS does not have low regret.

For evaluation, it will be more convincing if the regret bounds of the baseline algorithms are also shown explicitly in a table in the experiment section.

---

> ### Author Rebuttal · Authors · 2025-07-31
>
> We would like to thank the reviewer for their feedback.  Below we address the issues raised by the review.
>
> ## Weaknesses / Points for Clarification
>
> **Theoretical Inconsistency.**
> "Proposition C.1 and Proposition 1 are not consistent."
>
> **Our Response:**
> Proposition C.1 (appendix) is a more complete restatement of Proposition 1. There we clarify assumptions on the weights initialization distribution which were unfortunately misspecified in the main text.  In both the theoretical analysis and the code implementation, the weights were initialized following Gaussian Kaiming He initialization.  This scheme applies a variance which is inversely proportional to the previous layer's width, which corresponds to a variance of 1 for the first layer's weights and $1/M$ for the subsequent layers.  We will update this in the final version of the paper.
>
> **Experimental Clarity (NN Structure).**
> "It is unclear what neural network structure is used in the simulation of NOTS algorithm in the main paper, as the appendix specify the simulation is run under the single hidden layer NN."
>
> **Our Response:**
> For the experiments in the main paper, we used a multi-layer Fourier neural operator (FNO) architecture following the default settings from the examples in the `neuraloperator` package for their respective datasets.  In the appendix, however, we demonstrated that even with a single hidden layer and training only the weights of the readout layer, as suggested by our theoretical results, the proposed framework still achieves competitive performance.  One of the main objectives of this paper was to introduce a framework which is simple to implement with existing toolboxes and yet effective to tackle optimization problems in function spaces with little need for fine tuning.
>
> FNO settings for main paper experiments:
> * Darcy flow: 16 by 16 Fourier modes (2D spatial field), 64 hidden channels;
> * Shallow water: 32 by 32 Fourier models, 64 hidden channels.
>
> **Reproducibility.**
> "The simulation is not quite reproducible as some basic simulation setup is unclear.  Also, in some of the figures, the simulation has fewer iterations, which I think deserve an explanation."
>
> **Our Response:** The baselines that run for fewer iterations stopped due to the code crashing due to out-of-memory issues.  One important factor to note is that GP-based BO baselines need to keep all the data they observe to form, update and predict with their GP models.  As the inputs are quite high dimensional (e.g., the shallow water problem had 6144 input dimensions for each data point), and GPs need to keep their tensors at high numerical precision ( ideally 64-bit double-precision floats), GP-based algorithms can quickly run into memory issues.  In contrast, neural operator baselines only need to keep their weights for prediction after training.  The experiments code containing the full simulation setup will be released with the camera-ready version of the paper if accepted.  We will add these clarifications to the revised text.
>
> **Results Interpretation.**
> "In line 288, it says that 'STO-NTS thrives', but in Figure 2, STO-NTS does not have low regret."
>
> **Our Response:** Indeed, in Fig. 2, STO-NTS does not achieve low regret.  However, in line 288, we were referring to Fig. 1, where it achieves lower regret when compared to the other baselines, though still not as low as NOTS.  Hence, we will revise our claims on STO-NTS's performance.  In general, noting that STO-NTS uses a default fully-connected neural network architecture, as suggested by its original paper, without further adaptations to deal with high-dimensional spatial grid data, we believe that STO-NTS's poor performance is due to architectural limitations, which then lead to a need for higher amounts of data (i.e., more iterations) for its performance to become significantly better than the other baselines.  We will revise our statements on STO-NTS.
>
> **Suggestion on Baselines.**
> "it will be more convincing if the regret bounds of the baseline algorithms are also shown explicitly in a table in the experiment section."
>
> **Our Response:** We will add a table to the revised version of the paper.
>
> |Problem | RS                | BO                | NOTS              | BFO               | GP-TS              | STO-NTS            |
> |--------|-------------------|-------------------|-------------------|-------------------|-------------------|-------------------|
> | Darcy flow - rates  | 0.872 +/- 0.022 | 0.703 +/- 0.045 | **0.012 +/- 0.001** | 0.788 +/- 0.066 | 0.674 +/- 0.050 | 0.068 +/- 0.002 |
> | Darcy flow - potential | 0.456 +/- 0.012 | 0.240 +/- 0.031 | **0.099 +/- 0.060** | 0.462 +/- 0.062 | 0.130 +/- 0.021 | 0.4 +/- |
> | Darcy flow - pressure  | 0.077 +/- 0.001 | 0.047 +/- 0.001 | **0.012 +/- 0.001** | 0.078 +/- 0.006 | 0.038 +/- 0.004 | 0.068 +/- 0.002 |
> | Shallow water - inversion  | 4.632 +/- 0.876 | 1.639 +/- 0.532 | **0.134 +/- 0.043** | 3.076 +/- 0.886 | 1.942 +/- 0.502 | 2.329 +/- 0.800 |
>
> ## Questions
>
> **Question 1: "Good Flow Potential" Figure.**
> "The good flow potential figure also has open porus in the upper right and lower right corner.  Can the authors provide some explanations to it?"
>
> **Our Response:**
> The good flow potential figure was selected from 1,000 input-output pairs we have generated in the experiment;  therefore, the range of the functional was bounded by this pool and the global optimal value is not necessarily 0 which would ideally be achieved by closed pores.  The functional for this best candidate is -1.17 as opposed to the worst functional of -2.43, and indeed the flow rate of this figure was contributed by the open pores in the upper right and lower right corner.
>
> **Question 2: Rephrasing Lines 195-196.**
> "Could you rephrase line 195 to line 196. It is hard to understand."
>
> **Our Response:** We thank the reviewer for pointing this out.  We will rephrase it to: ``where the expectation is over the function space prior for $f$ and all other random variables involved in the decision-making process.'' We simply meant that the expectation in Bayesian regret is a \emph{total} expectation, involving all underlying random variables.
>
> ### Formatting Concerns
>
> **Typo (Line 230).**
> "Line 230: typo. deadline -> baseline"
>
> **Our Response:** Thanks for noticing that.  We will fix the typo in the revision.
>
> **Unclear Reference (Line 194).**
> "Line 194: the authors use the wording 'as previously discussed' without specifying the exact location where the result was given before.  This makes the paper hard to follow."
>
> **Our Response:** We unfortunately were unable to find the text the reviewer mentioned in line 194. The closest match was ``As previously described..." in line 289, which indeed can be somewhat unclear. It should be referring to Sec. 6.2 where we briefly specify the Darcy flow problem. More complete details about that problem's settings can be found in the appendix. We will revise this and other potentially unclear references in the text and make sure we specify section numbers in the main text and the appendix when referring to them.

---

> > ### Comment · Reviewer_Gc9e · 2025-08-02
> > **Response to unclear reference (line 194)**
> >
> > I was referring to line 194 in the appendix. Sorry for the confusion. And in general, it is a good practice to have clear references. I am glad that the authors found some other similar cases. Thanks!

---

> > > ### Author Response · Authors · 2025-08-04
> > > **Further clarification**
> > >
> > > Thanks for the clarification! Line 194 in the appendix was referring to the discussion about the trained neural operator approximately matching a GP posterior sample when considering the infinite-width limit. Such discussion is more clearly expanded in Sec. A, particularly at the end of Sec. A.2 (NTK). We will expand the proof to include an explicit reference and make the discussion more precise.

---

> > ### Comment · Reviewer_Gc9e · 2025-08-02
> > **Question on the baseline table**
> >
> > Can the authors explain what the numbers in the table are?
> >
> > Also, the baseline regret bounds I mentioned are something like the one in Proposition 2 (Eq.18). Can the authors summarize the bounds of all the baseline algorithms in a table？
> >
> > Thanks!

---

> ### Author Response · Authors · 2025-08-04
> **Baseline regret bounds**
>
> Regarding the table and the question on regret bounds for the algorithmic baselines, we have the following points.
> * The numbers in the table correspond to the average regret (i.e., $R_T / T$) of each algorithm at the end of their run.
> * Unfortunately, not all of the baseline methods have readily available theoretical bounds for their Bayesian expected cumulative regret.
>     - A trivial bound for random search (RS) is $\mathcal{O}(T)$, given that query points are chosen at random.
>     - Frequentist bounds are available for BFO, but we are unaware of bounds on its expected cumulative regret.
>     - Regret bounds for expected improvement, which was used for our BO baseline, in the Bayesian setting depend on rescaling the posterior variance or using the maximum of a GP sample path (see Takeno et al., 2025, below), which do not correspond to its standard setting as commonly found in practical applications and used in our experiments.
>     - Bayesian regret bounds are also unavailable for neural Thompson sampling and its variants, including STO-NTS, as they consider the frequentist setting, which again requires proper rescaling of an exploration term.
>
> The only directly comparable non-trivial regret bounds are, therefore, those of GP-TS, which are also $\mathcal{O}(\sqrt{T\gamma_T})$ in the Bayesian setting [38, main paper], where $\gamma_T$ here corresponds to the maximum information gain of the scalar-valued GP used by the algorithm.
>
> References
> * Takeno, S., Inatsu, Y., Karasuyama, M., & Takeuchi, I. (2025). Regret Analysis of Posterior Sampling-Based Expected Improvement for Bayesian Optimization. arXiv e-prints

---

> > ### Comment · Reviewer_Gc9e · 2025-08-04
> >
> > Thanks for the response. You have addressed my concerns. I will increase the score to 4.

---

> > > ### Author Response · Authors · 2025-08-09
> > >
> > > We are glad our response has addressed Reviewer Gc9e's concerns. We will make sure to incorporate the Reviewer's  constructive feedback into our paper's revision. Thanks again for your valuable insights and comments

---

### Note · Authors · 2025-08-12

We thank all reviewers for their constructive feedback. The main concerns (clarity of motivation, role of Thompson sampling, justification of optimization functionals, theoretical assumptions, and experimental scope) were addressed in detail in our rebuttal and discussion. Reviewers Gc9e and 9PFn confirmed their concerns were resolved and increased their scores; Reviewer 28oU maintained a borderline accept, noting limited domain familiarity; and Reviewer aMDd provided positive feedback with no further objections.

We clarified the framework’s goal, i.e., optimizing problem-specific functionals via neural operator surrogates within Thompson sampling, and explained its scalability and advantages over GP-based methods in high-dimensional functional settings. We detailed search space configuration, functional choices, and regret computation, and expanded explanations of theory, baselines, and experimental results. Reviewers recognized the novelty of extending Thompson sampling to function spaces with convergence guarantees, the practicality of the sample-then-optimize strategy, and the strong empirical performance on PDE-based benchmarks.

We believe the paper now presents a clear, rigorous, and impactful contribution at the intersection of neural operators, bandits, and functional optimization.

---

### Decision · Program_Chairs · 2025-09-17

**Decision:**

Accept (poster)

**Comment:**

**Summary:** The paper proposed the Neural Operator Thompson Sampling (NOTS) to solve optimization problems in function space. The specific optimization objective is the output of a known functional of an unknown operator. The paper presents a regret bound derived from an existing result on Gaussian process Thompson sampling, assuming a linear functional and a single-hidden-layer neural network. Experimental results on 2D Darcy flow and a shallow water model are presented to show the advantage of the proposed algorithm.

**Strengths:**
- Well-written paper with a good related work section and a clear contribution
- Novel combination of Thompson sampling with neural operators
- Theoretical guarantees in the infinite-width limit
- Good experimental evaluation.

**Weakness:**
- Computational cost of having to retrain the network.

**Decision and Suggested Changes:** The reviewers agree that the paper is well-written, and its contributions merit acceptance. After carefully reading the rebuttal/discussion, I tend to agree. Please incorporate the following comments in the final version of the paper. In particular, addressing the following concerns will help strengthen the current version of the paper:
- Clarify the theoretical inconsistency and add the experimental details/regret results for baseline algorithms (in response to Rev. Gc9e)
- Quantify and explicitly mention the computational cost to retrain (Rev. 28oU)
- In order to reduce the computational cost, investigate using SGLD to generate samples without complete retraining.
- Include an overview for TS in the background section (Rev. 9PFn)
- Include a justification as to why GPs cannot be directly used instead of neural networks (Rev. aMDd)